# Transforming Pipelines into Digital Platforms: An Illustrative Case Study Transforming a Traditional Pipeline Business Model in the Standardization Industry into a Digital Platform

**Davis Adedayo Eisape** 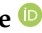

Chair of Innovation Economics at the Institute for Technology and Management, Faculty of Economics and Management, Technical University of Berlin, 10623 Berlin, Germany; davis.eisape@outlook.de

**Abstract:** For many, digital transformation is the new normal. However, in particular, pipeline businesses in traditional industries, such as standard-setting organizations (SSOs), are reluctant to radically rethink their business models, as they have often successfully prevailed for decades. The literature shows that there is a great deal of theory to be found on digital transformation, but a practical and, at the same time, scientific approach is yet missing. Following the design science framework, this paper introduces a two-step approach to transform a pipeline business model into a digital standardization platform. This is achieved by mapping the incumbent pipeline business mode146l and its ecosystem with the Platform Business Model Canvas introduced by Eisape. The representation of the current ecosystem is then digitally transformed according to the three key transformation points introduced by Alstyne et al., shifting the current ecosystem into the digital realm. The illustrative case study on DIN e.V. (the German SSOs) demonstrates the new methodology and its suitability for real applications. The result is a platform business model for a digital standardization platform, which, compared via an index to the traditional business model, has the potential to disrupt the entire standardization industry.

**Keywords:** digital transformation; platform business model canvas; ecosystem; standardization industry; DIN e.V.; case study

## 1. Introduction

Open innovation mainly refers to opening up the innovation process to knowledge and input from outside the innovating organization [1]. Open innovation is a paradigm that assumes that companies can and should exploit external, as well as internal, ideas [2]. As a result, the boundaries between a company and its ecosystem become increasingly blurred [3]. Innovative ideas can, thus, easily travel internally and externally, so that open innovation can also lead to innovative business models, among other things [4]. Business models can be understood as a broader concept that also refers to the overall logic by which innovations are commercialized [5]. As such, open innovation can result in all sorts of business models that have a cooperative, collaborative, or co-creative nature [6]. If a company based on a pipeline business model opens up its innovation process in such a way that it not only exchanges new ideas with external actors or receives them from external actors [7], but also develops a business model in which it acts as an intermediary for the exchange of ideas, services, products, technology, know-how, and other innovations between various external actors, then we are in the area of platform business models [8]. Gassmann et al. state that open business models focus on the integration of actors outside the company, that cooperative collaboration with partners is sought, and that the value chains and goals of the participating companies are matched [9].

Thus, platform business models basically embody the open innovation and open business model principles: externals create and demand value that is matched and leveraged for added value on a platform [10].

The digital transformation of a traditional business model has the potential to transform it into a disruptive force in its area of business [11,12]. In particular, the business models of digital platforms have turned entire industries around [13–16]. Uber, eBay, Airbnb, etc., are just a few very famous examples. Therefore, businesses in every industry are facing the challenge of innovating in order to remain relevant [17–19].

Indeed, markets and companies have been confronted with the effects of digitization for decades [20–24]. The increasing availability of computing power since the 1960s; the automation of workflows, processes, and workplaces in both manufacturing and administration; the commercial use of the internet since the mid-1990s; and the spread of increasingly intelligent mobile devices since 2005 have presented and continue to present companies with digital challenges [25–28]. Digitization has helped new business models to evolve. Vehicle manufacturers become multimodal transportation providers [29,30]. Manufacturers of analog products are expanding their range of services to include 24/7 service, lifetime function guarantees, and predictive maintenance intervals [31–34]. Meanwhile, 3D printing service providers are turning traditional development and production processes upside down, with rapid prototyping and rapid manufacturing [35–37]. Temporary employment agencies are automating the screening of candidates with digital speech analytics to increase the throughput of suitable candidates and control costs [38,39]. These are just a few examples of the digitization of business models.

However, digital transformation is fundamentally different from the mechanisms outlined above [40]: whereas, in the past, only individual industries, companies, and process steps within the company, as well as the variety of products and services, were affected by digitization, today, no industry, no company, and ultimately hardly any product or service remain unaffected by the effects of digital transformation [41]. Furthermore, it is not only individual elements of the value chain that are affected, but the entire value chain of a company—from the business model to the entire business ecosystem, including partners, customers, and other stakeholders [42]. This is true especially for digital platform business models that emerge in the wake of increasing digitization and ubiquitous connectivity, and disrupt pipeline ecosystems [43]. For example, brick-and-mortar retail and ordering via mail had to face the competition of e-commerce; analog cameras were basically forced out of the market, and the music industry, tourism industry, and printing industry, as well as publishing houses and newspapers, were subject to far-reaching industry upheavals [44]. Banks are facing a growing number of FinTech businesses that are claiming every single business process of banks for their own [45–47]. Against the backdrop of digital transformation, this development will continue to accelerate considerably, and companies need to be able to anticipate possible disruptive changes in their industries [48–50].

Jovanovic et al. introduce a holistic framework that helps to understand the core dimension in the development of so-called industrial digital platforms [51,52]. Like many other contributions in the literature, this framework offers a rich understanding on how to develop a digital platform, but misses out the "why". The "why" implies that companies have an excellent strategic understanding of what a platform business model in their business context may look like, before choosing a framework approach to develop one. More so, companies that successfully operate in traditional industries have been able to rely on the stable and familiar framework conditions of their industry for a long time. They shy away from radical steps regarding their core business models, as it means that they need to take risky steps with strong uncertainty about the probability of success [53].

To remain competitive, pipeline businesses need to be able to anticipate and develop a conceptual understanding of the potential a platform business model can unfold in their industry, and develop a strategic position towards the possible emergence of such. Bonina et al. distinguish between transaction and innovation platforms, where the former can be characterized as a digital multi-sided marketplace, and the latter as a technological facilitating space that allows contributors to offer or create further modules and complementors to use provided modules [54]. Eisape argues that all types of platforms, in essence, offer a digital space where contributors offer some sort of information, technology, goods, or

services, and consumers search, use, buy, and/or respond to the available offer, whereas so-called partners offer modules, information, and/or services that are core for the digital space to function [55]. However, an incumbent pipeline business should develop a strategic position on whether to be the facilitator of a possible digital platform, or position itself as a contributor, consumer, or partner [56]. Tools and roadmaps that can help to develop a possible platform business model have emerged [56–67], but often remain in theory and focus on the execution [68–70].

The scope of this paper is to introduce a business model transformation approach that helps develop and anticipate a conceptual platform business model on the basis of an existing pipeline business model and its core ecosystem. Specifically, within the design science framework, this paper uses the three key transformation points according to Alstyne et al. [56], and the Platform Business Model Canvas according to Eisape [55,71], to create a two-step approach for the digital transformation of an existing pipeline business model into a digital platform business model. This new approach is then tested for its adequacy using examples from the music and standardization industry.

This paper refers to open innovation, as it is a holistic approach to innovation management that systematically fosters and explores a broad range of internal and external innovation opportunities at the business model level, deliberately linking this exploration to the capabilities and resources of the business, and broadly leveraging these opportunities across multiple channels [72].

## 2. Literature Review

### 2.1. Transforming from a Pipeline to a Digital Platform in Two Basic Steps

Hanelt et al. state that "changing from a pipeline to a platform business model is particularly difficult. Whereas managers in traditional product companies have learned to concentrate on internal excellence and capabilities, they must now increasingly adopt an external focus, open interfaces, and enable value-creating interactions across boundaries with and among external players" [73]. As this statement implies, the transformation from a pipeline business model to a digital platform is only possible through the understanding of the current business's ecosystem. It is not enough to digitize the internal processes, products, and services of the pipeline value chain, but pipeline businesses have to understand their value creation contribution in an economic community that produces goods and services of value for customers who are themselves members of the ecosystem. Among the member entities are also suppliers, main producers, competitors, and other stakeholders [74]. Digital transformation is not just about the company's own value chain, but it affects the entire ecosystem [75–77]. Thus, positioning one's own pipeline value chain in the relevant ecosystem and shifting this ecosystem into the digital space at the business model level can anticipate the digital transformation to some extent and provide space for new potential [76–79]. This allows for the derivation of a two-step approach for transforming pipeline business models into digital platform business models (see Figure 1). The first step is to map the pipeline business model within its core ecosystem, and the second step is to "shift" the mapped ecosystem into the digital realm.

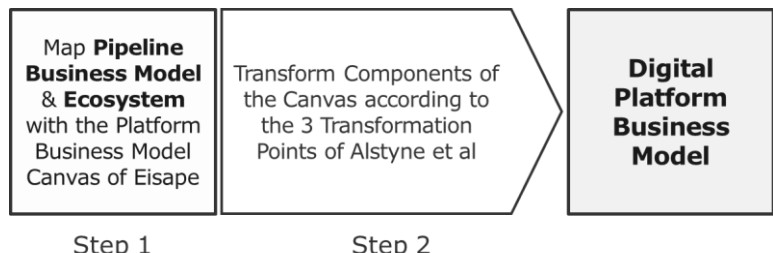

**Figure 1.** From a pipeline and its ecosystem to a digital platform business model in two steps.

### 2.2. Mapping the Pipeline Business Model and Its Core Ecosystem (Step 1)

In the business world, the term ecosystem is often defined in a reductionist way to include products, services, and companies that work together to deliver value to the customer.

> "An economic community supported by a foundation of interacting organizations and individuals—the organisms of the business world. The economic community produces goods and services of value to customers, who are themselves members of the ecosystem. The member organisms also include suppliers, lead producers, competitors, and other stakeholders. Over time, they coevolve their capabilities and roles, and tend to align themselves with the directions set by one or more central companies. Those companies holding leadership roles may change over time, but the function of ecosystem leader is valued by the community because it enables members to move toward shared visions to align their investments, and to find mutually supportive roles." [73].

James F. Moore, in his famous paper published in 1993, made the comparison between natural ecosystems and business ecosystems. Entities (living or legal), resources, coexistence, and interactions characterize both ecosystems [73]. Nachira, in 2002, referred to digital business ecosystems, where software enables a large number of interactions among users and services within and between organizations. Nachira explains digital business ecosystems as analogous to natural ecosystems, as something self-organizing and adaptive, and that they are digital environments inhabited by digital species [74,75]. A definition for "business ecosystem" was also developed by Iansiti and Levien [76,77]. They define business ecosystems as "loose networks of suppliers, distributors, outsourcing firms, manufacturers of related products and services, technology providers, and a variety of other organizations". Digital platforms are new ecosystems composed of platform participants [80]. These participants are connected by digital networks, and use the resources of the platforms, analogous to a natural or a business ecosystem. Platforms are characterized by interdependent and interactive relationships among participants [79,81]. Their key assets are interactions and information. This is the basis for the competitive advantage and value that platforms potentially create [56]. To summarize, business ecosystems describe how entities interact and coexist, just as digital platforms give entities the space for interaction and value transactions [80]. Again, one side cannot exist without the other side, and if one side evolves, this has a direct impact on the other side. Due to the similarity in essence between platforms and business ecosystems, the value propositions, transactions, and resources between the actors of a business ecosystem in its core can be understood as a platform business model.

Eisape's Platform Business Model Canvas is a method to visualize and describe the core logic of a platform business model [55], as shown in Figure 2. The components of the canvas do not merely map the actors of an ecosystem, but clearly depict the value proposition of actors, the transactions between actors, resources, and activities that need to be introduced to the platform by actors and the governance framework for a "healthy" ecosystem [55,71]. This helps to depict the business model of a pipeline business and its role within its ecosystem in coexistence with the other actors. Understanding the needs and expectations of platform participants is crucial to the success of the platform in order to effectively match them [71]. To achieve this, the Platform Business Model Canvas addresses the core logic from four perspectives. Thus, the Platform Business Model Canvas helps to facilitate the understanding and alignment of coexisting value propositions and activities [55]. In doing so, it considers all four dimensions sufficiently to develop an innovative and transactional platform [55]. The Platform Business Model Canvas consists of eleven different components and four perspectives [55].

Core Value Unit: The core value unit describes the technical unit that is created and consumed. It is the core of each platform and the connection point between all platform users.

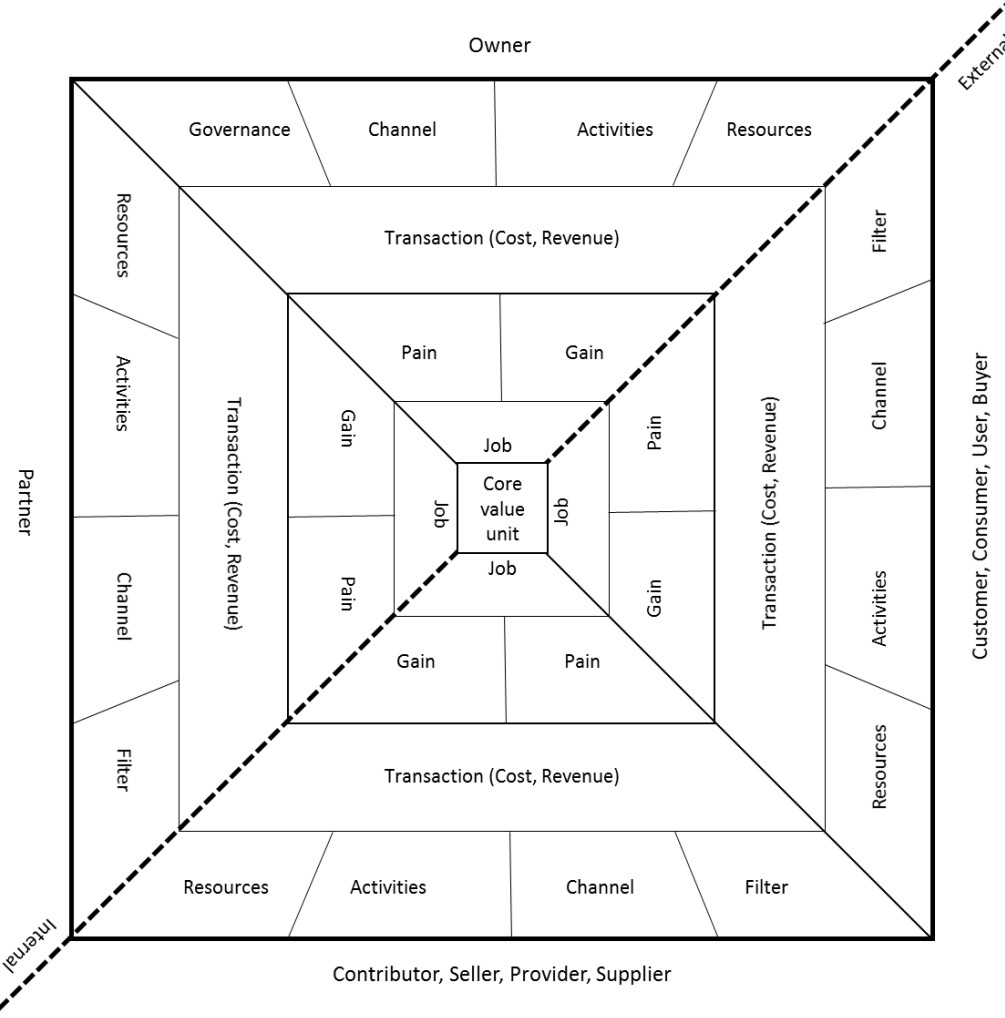

**Figure 2.** The platform business model canvas, by Eisape [55].

Job, Pain, Gain: The goal of platform users is to solve a problem or address a need. To make this possible, the platform needs to match complementary interests. Gain describes the benefits and motivation that come with meeting the demand or need. Pain, on the other hand, describes hurdles and problems that could arise while fulfilling the job.

Transactions: Transactions represent cost and revenue streams between the participants, and are always to be understood bidirectionally. Transactions refer to complementary corresponding interests in which two needs are met through the exchange of goods, information, services, or financial resources.

Key Activities: A representation of the key activities of all active participants is necessary to use the platform and, overall, keep the platform functional.

Key Resources: Active participation on a platform comes with different resources that need to be introduced by the users. Examples of such a resource could be data, time, knowhow, money, cost for personnel, hardware, etc. Key resources are the required inputs for the platform to function.

Access/Promotion Channels: Potential participants of a platform are addressed through promotion channels. Participants on the platform access the platform through specific access channels, which may vary between participants.

Filter: The use of filters is an important means of keeping platforms stable and functional. Filters can be used to ensure that the right people have access to the platform. The use of filters represents a trade-off between the quality and quantity of platform

participants because they enable barrier regulations, as well as the coordination of those who have access.

Governance: Good governance of the platform requires certain rules. These rules ensure that a healthy environment is created, and innovation and transactions are encouraged.

Consumers: The term may vary depending on the platform. It can be referred to as user, buyer, consumer, implementer, customer, etc.

Contributors: The designation can vary depending on the platform. These platform participants can be sellers, providers, suppliers, contributors, etc. They offer something on a platform and are essential to the transactions.

Key Partners: The success of a platform is often dependent on the technology, expertise, resources, and processes used by key partners to add value to the platform. Key partners, therefore, facilitate activities or increase the effectiveness of the platform. Furthermore, they can serve to share the potential risks of the platform.

Owner: The owner operates the platform and manages all activities on the platform, with the goal of growing the ecosystem community.

*2.3. Digital Transformation of the Ecosystem (Step 2)*

Digital transformation of businesses is nowadays considered to be the new normal [82–84]. Nevertheless, there is currently no generally accepted definition for the term digital transformation, but there are many attempts to define it [41,85–87]. Mergel et al. define digital transformation quite broadly as:

> "a holistic effort to revise core processes and services [ . . . ] beyond the traditional digitization efforts. It evolves along a continuum of transition from analog to digital to a full stack review of [products, services], current processes, and user needs and results in a complete revision of the existing and the creation of new digital services. The outcome of digital transformation efforts focuses among others on the satisfaction of user needs, new forms of service delivery, and the expansion of the user base." [61], p. 12.

This paper follows this definition, as it helps to describe the approach adopted by businesses that wish to evolve from a pipeline business model into a digital and ecosystem-based business model.

2.3.1. Digital Transformation According to Schallmo et al.

An often cited approach for digital transformation can be found in the works of Schallmo et al. [88,89]. Schallmo et al. propose a five-step roadmap for digitally transforming existing business models. Although the five steps introduced in 2017 and 2019 vary in wording and detail, the basic concept is the same.

In the first step, the existing business model is analyzed in detail, with regard to the value chain, existing digital contact points with customers, internal and external stakeholders, internal resources, the various target groups, as well as customer requirements and needs. The second step of the digital transformation involves defining and prioritizing the corporate goals into a "SMART" formulation. "SMART" is an abbreviation that stands for Specific, Measurable, Achievable, Realistic, Timed. Finally, the priority of the goals serves to bring about a decision as to the order in which new digital business models are to be tackled. The third step involves evaluating which opportunities arise for the company as a result of digital transformation, and which niche it can occupy with realistic effort and the available resources. The drivers of digital transformation are very diverse: they range from the generation of digital data, to automation, to networking and digital customer access. In the fourth step, a new business model is implemented, preferably employing agile project management methods to be able to react promptly to new requirements from the market and customers. In the fifth and final step of digital transformation, the new digital customer experience is closely examined. Every aspect of the newly created customer contact is analyzed and optimized. In addition, the new service is fully integrated into the company's

value chain. With the analysis of the feedback, the expanded business model is optimized in further iterations.

This approach is suitable for digitizing and digitalizing business models. Digitization applies to the internal optimization of processes (e.g., work automation, digital communication, and paper minimization), and leads to cost reductions. On the other hand, digitalization is a strategy or process that goes beyond the implementation of technology, and implies a deeper, central change in the entire business model and the transformation of operations. Following Schallmo et al.'s approach can, through various iterations and transformational steps, eventually transform a pipeline business model into a platform business model. Thus, the transformation of a pipeline business model into a digital platform is one possible outcome that eventually results from many iteration cycles.

### 2.3.2. Digital Transformation According to Alstyne et al.

The literature offers another transformational approach that specifically focuses on pipelines and platforms. According to Alstyne, Parker, and Choudary, the transformation from a pipeline business model to a platform business model requires three key transformation points [56], which are very much in alignment with Eisape's concept of "outsourcing dimensions of control" when shifting the perspectives from a pipeline business model to a platform business model [55]:

From Resource Control to Resource Orchestration: The resource-based view of competition states that companies gain advantages by controlling scarce and valuable assets. In a pipeline world, these include tangible assets, such as mines and real estate, and intangible assets, such as intellectual property. In platforms, the assets that are difficult to copy are the community and the resources that its members own and contribute, be they spaces or cars or ideas and information. In other words, the network of producers and consumers is the most important asset [56].

Here, the key transformation shift very much relates to the outsourcing of control over key activities, resources, and partners, which means "[ . . . ] to place the handling of key infrastructure into the hands of external partners. In this case, a business stands before the challenge to induce key partners to offer key product parts, processes, resources, technologies, know-how and activities to its business activities" [71], p. 94.

From Internal Optimization to External Interaction: Pipeline companies organize their internal labor and resources to create value by optimizing an entire chain of product activities, from material sourcing to sales and service. Platforms create value by facilitating interactions between external producers and consumers. Due to this external focus, they often even save variable production costs. The focus shifts from dictating processes to convincing participants, and ecosystem management becomes an essential skill [56].

This key transformation shift very much relates to the outsourcing of control over product/service creation, which means "[ . . . ] to place the production of goods or the provision of services into external hands" [71], p. 94.

From Focusing on Customer Value to Focusing on Ecosystem Value: Pipelines seek to maximize the life cycle value of individual consumers of products and services that are effectively at the end of a linear process. In contrast, platforms attempt to maximize the overall value of an expanding ecosystem in a circular, iterative, feedback-driven process. Sometimes, this requires subsidizing one type of consumer to attract another type of consumer [56].

This key transformation shift very much relates to the outsourcing of control over customer relationships, channels, and segments, which means "[ . . . ] that the control over the entire area of customer approach and customer management is outsourced, so that the "control" or better the creative initiative lies with the customers" [71], p. 94.

The approach from Alstyne et al. allows for the development of new platform business models that somewhat "fast-forward" the digitization iterations of Schallmo et al. At the same time, it is theoretical rather than practical, and lacks the clarity of Schallmo et al.'s roadmap. This paper aims to fill this gap, and offers a step-by-step-approach

for business model innovation that will transform pipeline business models into digital platform business models.

Before introducing the new approach, a short, retrospective digression considering the evolution of the music industry can help us to understand the underlying processes and the new proposed approach thereafter.

### 2.4. Learning from the Past: The Transformation of the Music Industry

The digital transformation of the music industry is a classic example to explain the steps for transformation at the business model level. The existing ecosystem of the music business, controlled by the five music majors (Sony, Warner, EMI, BMG, and Universal), was fundamentally transformed in the late 1990s [90]. The ringtone business, new decentralized and server-driven distribution systems (such as the file-sharing services KaZaA, WinMX, Gnutella, eDonkey), and online music stores (such as iTunes) expanded the existing music business ecosystem with new market players [90]. The pipeline business model dominating the industry ecosystem with a buyer–supplier relationship had changed into a more informal, collaborative, network-structured value web [90] (see Figure 3). Value webs are customer-centric ecosystems consisting of specialized, economically interrelated, but organizationally independent, institutions, yet collectively contributing to an integrated value creation or customer offering [90]. The value web broker (or, simply, value broker) takes on an orchestrating intermediary role here [90]. Companies such as Funtones, Musicwave, Smash Hits, Jamba, Last.fm, and Phunkytones appeared as aggregators and distributors, and pushed the majors out of the music distribution business [90].

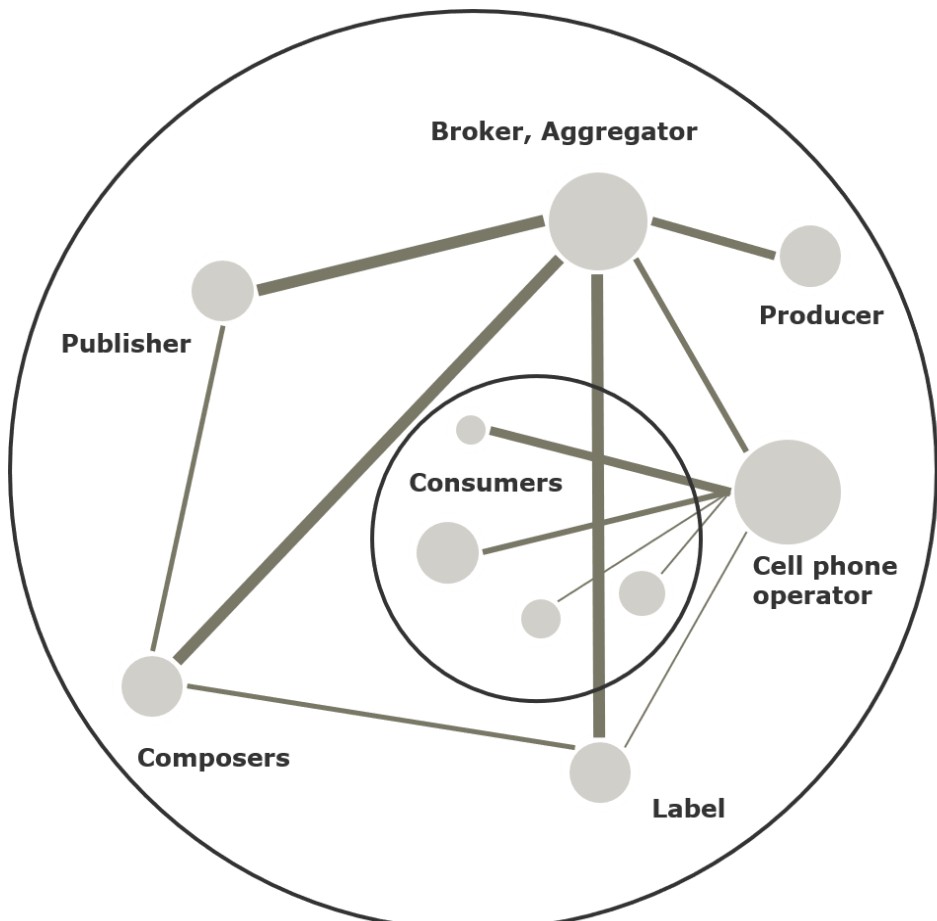

**Figure 3.** The music business ecosystem: music value web, according to Clement et al. [90].

The music majors found themselves in a broadened music business ecosystem, and needed several years to self-discover their place in the new environment [90]. It was primarily the inertia of the major music companies to recognize and respond to the digitalization of their ecosystem that left open entrepreneurial gaps into which young companies then ventured [90]. The music industry's attempts to retain control of the pipeline value chain in the networked ecosystem with digitization measures (e.g., super audio CDs, surround sound) were not very successful [90]. Today, we know that when platforms enter a pipeline-dominated business, the platforms almost always win [88,89].

Pipelines that want to tackle the emergence of platforms or transform into platforms themselves need to understand their value web (ecosystem) and the possibilities that the digitalization of the ecosystem presents for entrepreneurial ventures [90–92]. Thus, pipeline businesses need to prepare and take transformative steps towards a platform business model [93,94]. To demonstrate how these digitization/digitalization steps have changed an entire industry in the past, the music industry is a perfect illustrative example. Among other industries, the music industry helps to illustrate how, from its industrial revolution in the 1950s, it has evolved from pipeline business models to digital platforms.

### 2.4.1. The Transformation of the Music Industry According to Schallmo et al.

The starting point for the transformation was the digitization of the industry. With the invention of mp3 and the rise of the internet, interactions were possible at little cost. At the same time, the mere digitization of industries (such as the invention of the CD, or only the mp3, without the internet) did not trigger a transformation. It created higher efficiency, and reduced costs for the industry, as it simply substituted one product (the vinyl record) with another "digitized" product (CD, MP3), with little impact on the underlying "pipeline" business model. Transformation is the combination of digitization and radical innovation, resulting in new value creation. A record label that transforms from a record production company to a record store, to a lending library, and then shifts into the digital realm has radically transformed from a pipeline business model into a digital marketplace (platform business model), where consumers and producers are matched and create a community.

This development essentially reflects Schallmo et al.'s iterative approach of digitization and digitalization [88] (see Figure 4). This approach resonates well with rather conservative industries, which are still very successful, and, thus, have an inherent unwillingness to radically change their business models, as it means that they must take risky steps with strong uncertainty about the probability of success. Therefore, an iterative approach that stays close to the existing business model is seen as fitting. For decades, they have been successful with the approach of simply continuing to improve their existing products and services and creating digital variations. The music industry is a great example. The transformation from pipeline business models to platforms business models in the music industry took over 50 years to evolve. One may argue that businesses today do not have ample time to gradually innovate their business models and adapt to the ever-increasing innovation cycles. In addition, a strategy is needed to be able to have a position on possible platform business models, disrupting their industry.

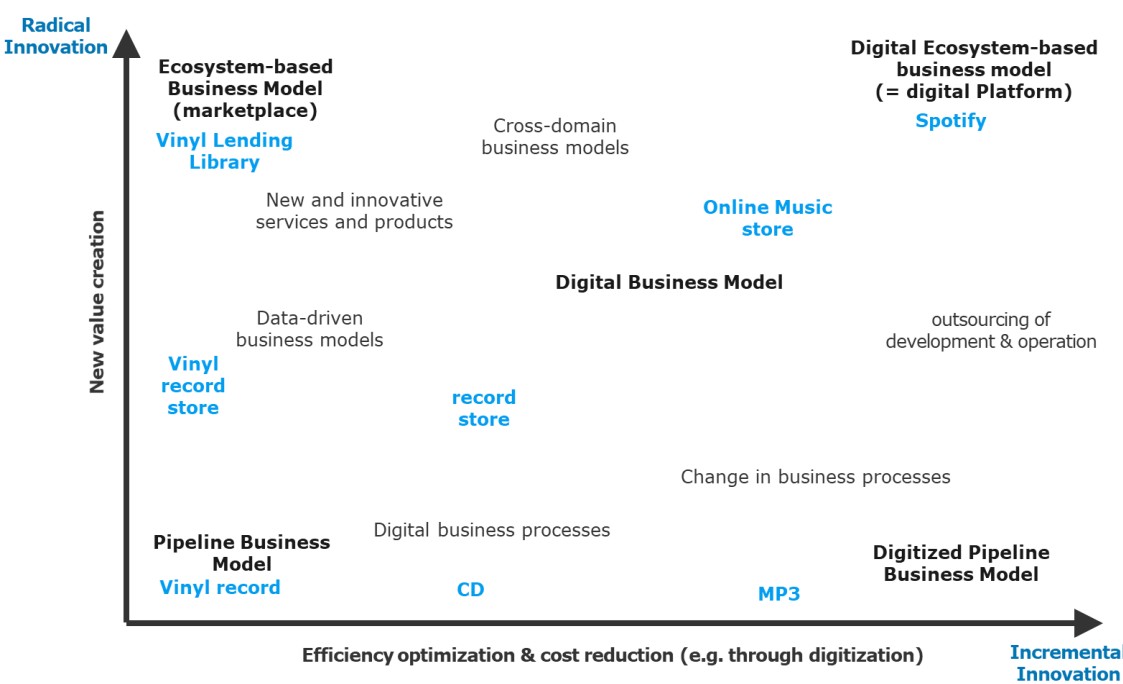

**Figure 4.** The path from pipeline business models to digital platform business models according to Schallmo et al. (adapted from Trapp et al. [95]).

2.4.2. The Transformation of the Music Industry According to Alstyne et al.

The digital transformation according to Alstyne et al. can also be demonstrated through the illustrative case study of the music industry (see Figure 5).

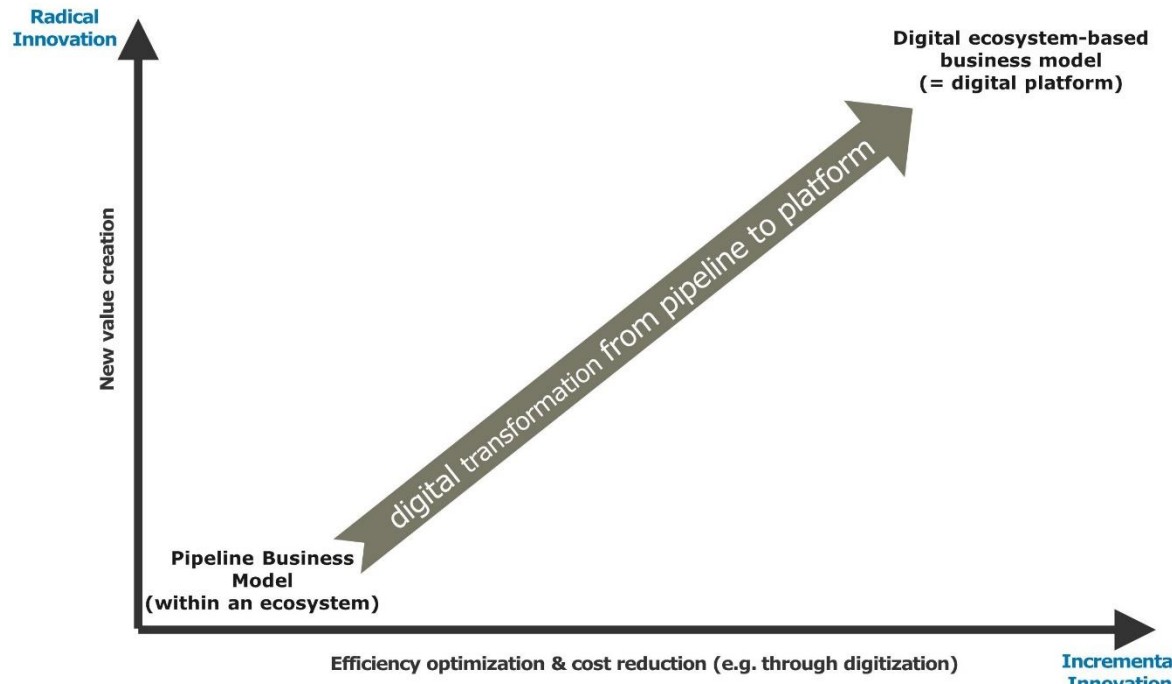

**Figure 5.** The path from pipeline business models to digital platform business models according to Alstyne et al. (adapted from Trapp et al. [95]).

Through the lens of Alstyne et al.'s first key transformation point, "from resource control to resource orchestration", the following shift in the music industry took place: in the past, the music industry was very successful in recording artists and then selling the

music to consumers. The pipeline business model was perfect, and the hunger for new talent, the signing of talent in exclusive contracts, and marketing to consumers were the success factors for many labels, producers, and publishers. In the new age, with the rise of platforms such as Spotify, and the almost abundant presence of artists and music, the asset that is difficult to copy is the community of musicians, producers, DJs, and consumers that are matched, which creates value for consumers and contributors and, thus, also for the platform [96].

Through the lens of Alstyne et al.'s second key transformation point, "from internal optimization to external interaction", the following shift in the music industry took place: in the past, record companies had to organize their internal labor and resources to create value by optimizing the entire chain of product activities, from scouting new talent to producing the music, and then selling and marketing it. In the new age, platforms such as Spotify create value by facilitating interactions between "external" producers (artists, producers, labels, etc.) and consumers. Due to this "external focus", Spotify does not even have to produce music or search for artists. The focus shifts from production processes to matching demand and offer, facilitating interactions among platform participants. The ecosystem management has replaced the product management.

Through the lens of Alstyne et al.'s third key transformation point, "from focusing on customer value to focusing on ecosystem value", the following shift in the music industry took place: in the past, record companies put a great deal of effort into maximizing the life cycle value of individual consumers, aiming at maximizing the amount of money that a customer contributed to the record company. For this, they created additional products besides the music itself, such as merch, fan products, posters, magazines, etc. All of these are effectively at the end of a linear production process. In contrast, in the new age, platforms such as Spotify attempt to maximize the overall value of their ecosystem in a circular, iterative, feedback-driven process. More music and better recommendations increase the number of songs that consumers listen to and how much time they spend on the platform. This convinces musicians and record labels to share more music on the platform. This, in return, attracts even more consumers, allowing the platform to grow its brand, reach, and value.

*2.5. The Platform Index*

To remain competitive, companies need to understand how to innovate or transform their pipeline business models into platform business models. To find the right platform business model, "[ . . . ] being able to compare platform business models (e.g., with the help of a platform business model canvas) is key" [97]. Eisape has introduced seven measurable key performance indicators that use a radar chart to compare two or more platform business models: "market share of consumer, market share of provider, amount of partnerships/degree of networkedness, brand value of owner, amount of core value units, amount of auxiliary value unit and the revenue value and diversity—all ranging from low to high in 5 steps" [97], p. 419.

For the direct comparison of two platform business model versions, without an objective baseline ([97], p. 425), the scale from 1 to 5 is retained in this work, and the meaning of the expressions is interpreted as follows:

- 1 means that the considered attribute on platform A is significantly worse than the same attribute on platform B. Platform B would then be 5.
- 2 means that the considered attribute at platform A is slightly worse than the same attribute at platform B. Platform B would then be at 4.
- 3 means that the considered attribute on platform A is roughly comparable to the same attribute on platform B. Platform B would then also be 3.
- 4 means that the considered attribute on platform A is slightly better than the same attribute on platform B. Platform B would then be at 2.
- 5 means that the considered attribute on platform A is significantly better than the same attribute on platform B. Platform B would then be 1.

With the radar diagram, both the characteristics of the seven evaluation criteria (revenue value and diversity, market share of consumer, market share of provider, amount of partnerships, brand value of owner, amount of core value unit, and amount of auxiliary value units [87]) and the area covered in the radar diagram help to compare two business models. The interpretation follows the logic wherein the larger the area covered, the higher the level of the characteristics and the potential for success in terms of a platform business model. Thus, the surface area quantifies the entirety of the characteristics ($a_x$). In relation to the maximum possible area size in the radar diagram ($A_{max}$), a platform index ($i_{Platform}$) is then calculated, which ranges between 0 and 1.

The formula is structured as follows:

$$i_{Platform} = sin(360°/7)/2 \cdot A_{max}\left(a_{revenue} \cdot a_{consumer} + a_{consumer} \cdot a_{provider} + a_{provider} \cdot a_{partner} + a_{partner} \cdot a_{brand} \right.$$
$$\left. + a_{brand} \cdot a_{core} + a_{core} \cdot a_{aux} + a_{aux} \cdot a_{revenue}\right)$$

$$a_x = \{1, 2, 3, 4, 5\}$$

$$A_{max} = 80.33$$

A critical aspect with regard to the comparison is the availability and choice of data [98,99]. Depending on which data are available and used, the result of the comparison may vary significantly. This could be a subject for further research to identify the right set of data for comparison for each KPI. In this paper, the data used come from the websites of the ecosystem players, and represent the latest or current status. This means that a platform business model is not compared with fictive data, but with current data, as if the data would apply to the platform today. Of course, this neglects possible effects (network effects, risks, dynamic competition, etc.) that would alter the data significantly. However, it offers a mutual ground to compare two platform business model concepts according to Eisape.

## 3. Materials and Methods

Following the design science framework (see Table 1), this paper will employ this methodology to build and evaluate constructs, a model, and an instantiation that help pipeline businesses to transform their business models into digital platforms. March and Smith presented design science as a scientific category to find solutions to real-world problems [55,97,100–102]. They defined four types of output with regard to design science: constructs, models, methods, and instantiations [103,104]. "Constructs" are a collection of terms used to describe and define artefacts and phenomena. "Models" are descriptions of situations, tasks, and artifacts involving constructs. "Methods" are target-oriented instructions for action with the integration of constructs and models. "Instantiations" are the practical implementation of methods and models for certain tasks [103,104]. March and Smith [100] proposed a structure with four-by-four cells [105]. The different cells have different objectives with different suitable research methods. A research project can cover several cells, but not necessarily all of them [105]. In terms of research activities, March and Smith identify build and evaluate as the two main topics in design science [105]. In parallel to these two research activities, March and Smith add the natural and social science pair, namely theorize and justify. Theorize refers to the construction of theories that explain how or why things happen. Justify refers to theory validation, and requires the collection of scientific evidence that supports or refutes the theory. These two research activities become relevant when a model is widely used, and certain effects and results are observed that need to be explained by a theory that is then justified [105]. This is not within the scope of this paper.

**Table 1.** Design science framework according to March and Smith.

| Research Activities | Research Output | | | |
|---|---|---|---|---|
| | Research Output | Model | Method | Instantiation |
| Build | Define necessary steps to digitally transform pipeline business models into digital platform business models (see Section 2.1) Method: Literature review | Define a transformation model (methodology) that will enable firms to transform pipeline business models into a digital platform business model (see Sections 2.2 and 2.3) Method: Literature review | | Visually transform a pipeline business model into a digital platform business model (see Sections 4.3–4.5) Method: Literature review, Platform Business Model Canvas |
| Evaluate | Test adequacy, Evaluate the adequacy of the steps via an illustrative case study (see Section 4.2) Method: Illustrative case study on DIN e.V. | Test adequacy, Evaluate the model's adequacy via an illustrative case study (see Section 4.3) Method: Illustrative case study on DIN e.V. | | Describe the new digital platform business model of the illustrative case study and compare it to the pipeline business model via an index (see Sections 4.4 and 4.5) Method: Illustrative case study on DIN e.V. |
| Theorize | | | | |
| Justify | | | | |

The methodologic steps taken in this paper, in accordance with the design science framework, cover the research activities Build and Evaluate, and result in the research output of Constructs, Model, and Instantiation. As already mentioned, the main objective of this paper is to Build an approach that will support a systematic methodology in transforming pipeline business models into digital platform business models. Therefore, Constructs and Model will develop the steps of the approach, as well as the transformation mechanism. Instantiation will visualize the components and the transformation mechanism, and compare the indicators of two platform ecosystems, both related to the DIN e.V. illustrative case study. The research activities Theorize and Justify, as well as the research output Method, are not part of this research. Testing the adequacy of the attributes and characteristics against reality will be performed through illustrative case studies. This paper uses the illustrative case study in accordance with Stratmann [106] to test and illustrate the adequacy of the steps, the transformation mechanism, and their instantiation. Jahn explains that the illustrative case study is appropriate to illustrate previously identified basic patterns, such as a model [107]. The methodological context put forward by Hevner, March, Park, and Ram underlines that the solution orientation of this paper is sufficient for a scientific work, as "(1) business needs motivate the development of validated artifacts that meet those needs, and [...] (2) the development of justified theories about these artifacts produces knowledge that can be added to the shared knowledge base of design scientists" [103,104].

Within the design science approach, the transformation starts with understanding the pipeline business model within its ecosystem. After developing the two-step transformation approach, to illustrate its adequacy, the standardization business model of DIN e.V. is presented in the context of its ecosystem, consisting of the four perspectives, Consumer, Producer, Partner, and Owner, showing the internal processes of all four perspectives and the integrated overall logic of value creation within the core ecosystem. The Platform Business Model Canvas introduced by Eisape [68,69] visualizes the logic of the value-creating ecosystem elements in analogy to a platform business model. The Platform Business Model Canvas, presented in this paper, has been prototyped and employed in several industry-related Master's and Bachelor's theses, where it has been used to model and develop platform business models with an array of modeler profiles and contexts [68,69].

In the second step, according to Alstyne et al., the transformation from a pipeline business model into a platform business model requires three key transformation points [56].

Each component, as well as the overall logic of the business model, and, thus, the interactions and roles of ecosystem stakeholders, are then revised along these three key transformation points. This can be performed by searching for hints in the literature, employing brainstorming techniques with experts, interviewing experts, or by other similar approaches, with regard to each field of the Platform Business Model Canvas and each transformation point. In this paper, the author uses the illustrative case study to solely demonstrate the model, by adapting ideas from websites and the literature. Therefore, the result has no claim for correctness and completeness. One may argue that even a well-discussed input by experts also offers no guarantee that the new business mode will be more successful.

Nevertheless, there are some factors that can help to indicate whether the new business model has higher theoretical potential compared to the old one. In the context of the transformation and, thus, the generation of the one or more platform business model variants, a comparability approach is needed, which helps to identify the best possible constellation. To identify the constellation with the greatest possible value creation potential and reach, the variants are compared using key performance indicators for successful platform business models [97]. Eisape introduced a methodology to compare two platform business models via key performance indicators in a radar diagram [97]. This work goes one step further and develops an index that reflects the area covered in Eisape's radar diagram, and, thus, expresses which ecosystem constellation has the most promising platform potential. The data for the comparison are taken from the literature and the company's websites. They are current data that do not add any network effects or risks. Thus, essentially, this paper compares the current pipeline business model in its ecosystem to a theoretical digital platform based on the same data.

## 4. Results

### 4.1. The Field of Standardization: The Case of DIN e.V.

The field of standardization is a conservative industry that, for over 100 years, has essentially managed to preserve the basic underlying business model. Nevertheless, standard-setting organizations (SSOs), due to their central role in their economies, often refer to themselves as being "platforms". The German institute for standardization states that "DIN, the German Institute for Standardization, is the independent platform for standardization in Germany and worldwide" [108]. Böhm and Eisape describe in their phase model of standardization, the de jure and de facto standardization [109]. Standardization organizations (such as DIN Deutsches Institut für Normung e.V.) set formal and recognized standards that follow the de jure path within the phase model ([108], p. 113) (see Figure 6). They are very much the point of contact for market demands on standardization activities (Standard Pull). Mandates from the regulatory level, but also their own initiatives, can also initiate standardization activities (Standard Push). In the ideation phase, initiators and experts from relevant stakeholders are brought together, and a first draft is prepared, evaluated, discussed, and released for a public commenting phase. This is followed by the specification phase, in which the experts finalize the standard according to the strict rules of the standardization organization and with the involvement of all interested parties. In the diffusion phase, the standard is published so that it can be used on the market. The extent to which the standard is accepted depends very much on how relevant, adequate, and early it is with regard to the needs of the market ([108], p. 118). Privately organized standards organizations generate revenue through the sale of standards; through membership fees paid by the experts, who can, thus, participate in standardization within the framework of thematically sorted expert committees (so-called technical committees); and through public funding to promote politically important projects, as in the case of DIN e.V., for example, with the early promotion of technical innovations [110].

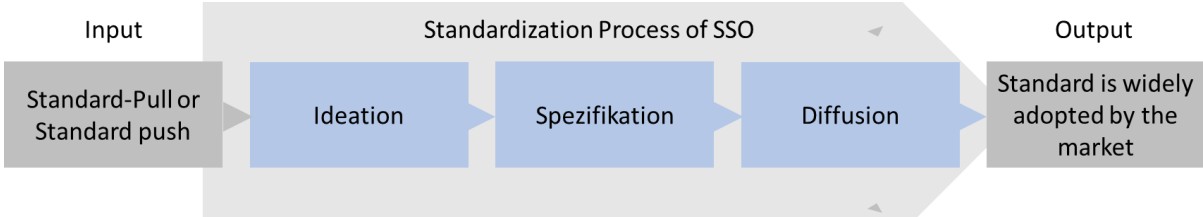

**Figure 6.** The pipeline business model of standard-setting organizations (SSO) (adopted from Boehm and Eisape, [109], p. 113).

Standard-setting organizations can be understood as service providers, in the sense of a linear so-called pipeline business model [97,110], p. 395, [111], since this is where value creation takes place from the producer to the consumer [97].

Of course, SSOs show great effort in digitizing their processes, services, and products, and even expanding their service and product portfolio through digitalization [112,113]. SSOs have shown in the past that they are able to adapt to changes in the paradigm. Shifting from mail to email, from offset printing to on-demand digital printing, and from standards in a PDF format to the XML format are just a few examples of digitization in the realm of standardization [114]. Streamlining and fostering collaboration in virtual standardization committees and offering standards as code are long-stated goals that are even pursued by dedicated programs [113–116]. In addition, due to convergence, increased innovation rates, and shorter product life cycles, users will be confronted with an increasing complexity of standards. As a result, the demands on standardization are changing more and more noticeably. The affected customer groups are demanding greater transparency in the standardization process and shorter lead times. Therefore, changes in the standardization policy framework cannot be ruled out [116]. Thus, SSOs, such as the German Standardization Organization, DIN e.V., understand that a reorientation is necessary [116].

### 4.2. The Pipeline Business Model and the Ecosystem of DIN e.V.

Mapping the pipeline business model and the ecosystem of DIN e.V. unto the platform business model canvas is performed by describing all components (see Table 2, see also Figure 7). The consumers in the logic of the platform business model canvas are large, small, medium, and micro-enterprises [116–118]. They are part of the DIN e.V. ecosystem, as they want to inform themselves, research around standardization topics, and buy standards- or license-related software to store and manage their standards centrally. They need to read and understand the standards, as well as apply them correctly. Documentation can help in ensuring effective implementation. Finally, consumers wish to monitor the standardization progress in areas of interest and keep up to date [117,119]. The challenge that they may face is that the quantity of standards is now poorly manageable and is sometimes difficult to read and to understand, which can promote "black box thinking" with consumers. Standards are sometimes not up to date; they show evidence of a lack of practical testing or inefficient work of standards committees. In the end, buying standards come with high costs for consumers [120], p. 95, [121]. On the positive side, for consumers of standards, they help to reduce trade barriers (internationally); increase the security of products and services; promote clarity about the properties of a product and service; help to define interfaces and compatibility requirements of products, systems, and services; promote an increase in demand for products and services through greater consumer confidence and acceptance; and allow for simplified order processing and procedures [122,123]. Consumers can access the website of Beuth Verlag, which is the publisher of DIN standards in Germany, to purchase a DIN standard online 24/7. It also possible to read through standards free of charge at 90 so-called info points [117,123,124]. With regard to consumer transactions, they pay a purchase price to Beuth Verlag and receive a copy of the standard, secondary literature or training, etc., in return [108]. Consumers have several channels through which

to access the ecosystem of DIN e.V. through the 90 info points; online applications such as Standards Ticker, Perinorm, Beuth e-NORM, and Beuth Standards Manager; the online portal for commenting drafted standards ("Norm-Entwurfs-Portal"); and, to some extent, the offline platform of the German Committee of Standards Users (ANP) [116,117,123–126]. Consumers need to invest financial resources in order to buy a copy of a technical standard, and internet access in order to be able to access the DIN ecosystem [127]. Key activities that need to be performed are to locate an info point and use the computer, search online and purchase, access "Norm-Entwurfs-Portal", or join ANP [124–126].

**Table 2.** Components of the ecosystem of DIN e.V.

| Component | Description of Component | Source |
|---|---|---|
| Consumer: | -Large enterprises<br>-SMES<br>-Micro-enterprise | [116–118] |
| Job for consumer: | -Inform<br>-Research<br>-Procure and license<br>-Store and manage centrally<br>-Read and understand<br>-Apply correctly<br>-Documentation<br>-Monitor and keep up-to-date | [117,119] |
| Problems for consumer: | -Quantity hardly manageable anymore<br>-Difficult to read and more practical<br>-"Black box thinking"<br>-Not up-to-date<br>-Inefficient work of standards committees<br>-Lack of practical testing<br>-High costs for the users | [120], p. 95, [121] |
| Gain for consumer: | -Reduction of trade barriers (internationally)<br>-Increased security<br>-Clarity about the properties of a product and service<br>-Definition of interfaces and compatibility requirements<br>-Demand through greater consumer confidence and acceptance<br>-Simplified order processing and procedures | [122,123] |
| Filter for consumer: | -Available online, 24/7<br>-Available free of charge at 90 info points | [117,123,124] |
| Transaction for consumer: | -Pay price for standards to Beuth<br>-Receive a copy of a standard, secondary literature, training, etc. | [118] |
| Channel for consumer: | -Info points<br>-Online<br>-Standards Ticker<br>-Perinorm<br>-Beuth e-NORM<br>-Beuth Standards Manager<br>-Platform of the German Committee of Standards Users (ANP)<br>-Draft standard portal for commenting | [116,117,123–126] |
| Key resources for consumer: | -Financial resources to buy a copy of a technical standard<br>-Internet access | [127] |
| Key activities for consumer: | -Locate an info point and use the computer<br>-Search online and buy<br>-Access "Norm-Entwurfs-Portal"<br>-Join ANP | [124–126] |

**Table 2.** *Cont.*

| Component | Description of Component | Source |
|---|---|---|
| Provider: | Experts from<br>-Industry<br>-Research<br>-Consumer side<br>-Public authorities | [127,128] |
| Job for provider: | -Active participation in the standardization committee<br>-Formulating standards<br>-Bringing in technical solutions | [129], p. 1 |
| Problems for provider: | -Time expenditure<br>-Costs<br>-Lengthy consensus process<br>-Conflicts of interest with other participants<br>-Rising travel costs<br>-Language barriers | [130] |
| Gain for provider: | -Shaping the content of standards<br>-Direct exchange with other interest groups<br>-Knowledge advantage<br>-Making own company known<br>-Influence, acceptance, and respect from others | [122,128] |
| Filter for provider: | -Only experts from legal entities<br>-Paid membership<br>-Participation in a standards committee | [129] |
| Transaction for provider: | -Pay money to DIN<br>-Get access to standards committees | [129] |
| Channel for provider: | -Norm-Entwurfs-Portal für Kommentierungen<br>-Livelink<br>-Standards committee<br>-ANP | [125,128] |
| Key resources for provider: | -Membership of a legal entity<br>-Contribution to costs<br>-Recognition of the rules of standardization work<br>-Direct costs (travel, personnel, material, training costs, etc.) | [128,131] |
| Key activities for provider: | -Authorization<br>-Granting of the copyright usage rights<br>-Declaration of confidentiality<br>-Get to know documents and research options<br>-Acquire knowledge of standardization | [129] |
| Partner: | Beuth Verlag | [132] |
| Job for partner: | -Distribution of standards<br>-Standards management tools<br>-Accompanying literature<br>-Training | [117,130,132,133] |
| Problems for partner: | -Always suitable for industry needs<br>-All national, European, and international norms, standards, and technical regulations<br>-Clarity and high quality in the search function | [117,132] |
| Gain for partner: | -Brand of DIN e.V.<br>-Offers from DIN e.V.<br>-Topics of DIN e.V.<br>-Experts of DIN e.V. | [118] |
| Filter for partner: | -Close connection<br>-Economic contribution to DIN | [118] |

**Table 2.** *Cont.*

| Component | Description of Component | Source |
|---|---|---|
| Transaction for partner: | -Receives money from buyers<br>-Gives copy of standards to buyer | |
| Channel for partner: | -Livelink | [134,135] |
| Key resources for partner: | -Employees<br>-Active authors<br>-Web shop<br>-Server | [118] |
| Key activities for partner: | -Publish standards<br>-Develop accompanying offers<br>-Technical literature for all important industries and professional groups<br>-Digitally formatted expert content<br>-Software solutions for standards management<br>-Further training via the DIN Academy | [118] |
| Owner: | DIN e.V. | [128] |
| Job for owner: | -Monitor and manage standardization process<br>-Coordinate cooperation<br>-Involve all interested parties<br>-Ensure compliance of processes<br>-Ensure quality of standards<br>-Prepare publication of the standards | [127,136] |
| Problems for owner: | -Mirroring trends, convergence of topics, and innovation cycles<br>-Reduce processing time<br>-Zero error tolerance | [115,137] |
| Gain for owner: | -The only recognized national standards organization<br>-Relevance and importance for the economy | [127,138] |
| Transaction for owner: | -Receives money from providers<br>-Gives project management and process management services to providers | [127] |
| Channels for promotion by the owner: | -Fairs<br>-DIN-Mitteilungen/DIN-Anzeiger<br>-Website<br>-Flyer<br>-Events<br>-Further training<br>-General meetings | [139–142] |
| Key resources for owner: | -Established and secure processes<br>-Specialized employees<br>-Secure document management system | [128] |
| Key activities for owner: | -Facilitation infrastructure<br>-Involvement of all interested parties<br>-Organization of the standards committees<br>-Standardization process management up to publication<br>-Identification of innovative topics and technologies<br>-Review of existing standards<br>-Ensuring uniformity of all technical standards | [143] |
| Governance by owner: | -Rules of DIN 820<br>-Arbitration proceedings | [143,144] |
| Core value unit: | Listing of a specification (standard) or standardization project, services, literature | [145–147] |

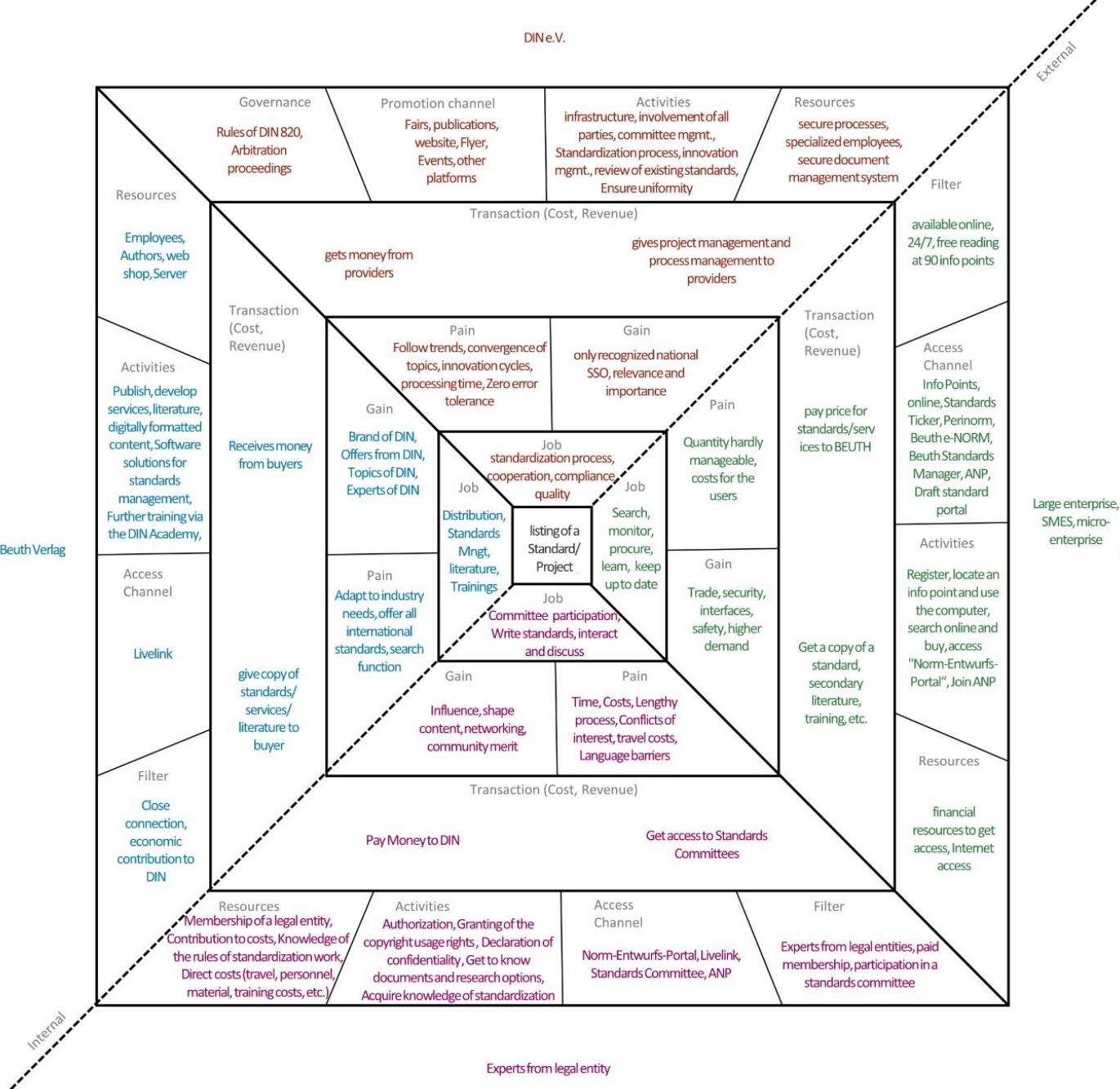

**Figure 7.** The ecosystem of DIN e.V. mapped via the Platform Business Model Canvas according to Eisape [55].

Providers, in the sense of the Platform Business Model Canvas, are experts from industry, research, the consumer side, or public authorities [127,128]. They approach the ecosystem, as they want to participate in a standardization committee or author standards by contributing technical solutions [129], p. 1. Their pains are time expenditure, membership and participation costs (e.g., rising traveling costs), lengthy consensus processes, conflicts of interest with other participants, and sometimes even language barriers [130]. On the other hand, the gain by shaping the content of standards, having direct exchange with other interest groups, gaining a knowledge advantage, making their own company known, and gaining influence, as well as acceptance and respect from others [122,128]. The ecosystem only gives access to experts from legal entities, which hold membership and participate in a standards committee [129]. Providers pay a membership fee to DIN

e.V. and obtain access to standards committees in return [129]. The channels of access are "Norm-Entwurfs-Portal", the online document management and exchange tool "Livelink", participation in a standards committee, and the ANP [125,128]. Resources that providers have to invest are membership of a legal entity, a financial contribution to DIN' s costs, the recognition of the rules of standardization work, and further direct costs (e.g., costs for travel, personnel, material, training) [128,131]. Mandatory activities to be able to participate within the ecosystem are subject to an authorization process, the granting of the copyright usage rights to DIN e.V., the signing of the declaration of confidentiality, learning about documents and research options, and acquiring knowledge on standardization processes [129].

A key partner for the ecosystem of DIN is Beuth Verlag [132]. Beuth offers the distribution of standards, various standards management tools, accompanying literature, and respective training [117,130,132,133]. Challenges for Beuth are to always have suitable offers for changing industry needs; to make available all national, European, and international standards and technical regulations; and to offer user-friendly clarity and high quality in the search function [117,132]. It benefits from the closeness to the brand, the offers, the topics, and the experts from DIN e.V. [118]. Beuth is DIN's only publisher, and has this unique position because it is a part of the DIN Group and makes a significant economic contribution to DIN's revenue [118]. With regard to transactions, Beuth receives money from buyers, and provides copies of standards, training, literature, etc., to buyers [118]. Beuth accesses the ecosystem through Livelink [134,135]. Key resources for the partner are employees, active authors, and the facilitation of a web shop on a server system [118]. Key activities of Beuth are to publish standards, develop accompanying offers, provide technical literature for all important industries and professional groups, digitally formatted expert content, software solutions for standards management, and to develop and promote training offers from the DIN Academy [118].

The owner of the DIN ecosystem is DIN e.V. [128]. DIN's job is to monitor and manage standardization processes, to coordinate cooperation, to involve all interested parties, to ensure compliance to processes, and to maintain high quality within all standards, as well as to prepare publications of the standards [127,136]. DIN e.V. faces challenges when it comes to mirroring trends in an adequate and timely manner, the convergence of topics and innovation cycles, reducing processing time, and enforcing a zero error tolerance [115,137]. As the only recognized national standards organization, it enjoys very high relevance and significance for the German economy [127,138]. With regard to transactions, DIN e.V. receives money from providers, and executes project management and process management services for providers in return [127]. DIN e.V. promotes its services through fairs, regular publications, e.g., "DIN-Mitteilungen" and "DIN-Anzeiger", its website, flyers, events, training, and general member meetings [139–142]. DIN e.V. invests key resources in facilitating established and secure processes, specialized employees, and a secure online document management and exchange system [128]. DIN's key activities center around the facilitation infrastructure, involvement of all interested parties, organization of the standards committees, standardization process management and publication preparation, identification of innovative topics and technologies, review of existing standards, and ensuring uniformity of all technical standards [143]. DIN e.V. has set up the binding standardization rules of DIN 820 and arbitration proceedings as governance for the ecosystem [143,144].

The core value unit (which is to be understood as the minimal technical common denominator of all four ecosystem perspectives) is the listing of a technical specification (standard) or a standardization project or related offers (literature, training, and information) [146–148].

### 4.3. Transformation of the Standardization Ecosystem towards a Platform Business Model

Transforming the pipeline business model of DIN e.V. in the context of its ecosystem into an online platform business model follows the three transformation points of Alstyne et al. mentioned above (please see Sections 2.3.2 and 2.4.2). The results are presented in the following three subchapters (also see Table 3).

**Table 3.** Transformation of the components according to Alstyne et al.

| Component from DIN e.V. and Current Ecosystem | 1. Transformation: From "Resource Control" to "Resource Orchestration" | 2. Transformation: From "Internal Optimization" to "External Interaction" | 3. Transformation: From "Customer Value" to "Ecosystem Value" | New Component in Digital Platform Business Model for Standardization |
|---|---|---|---|---|
| Consumer: | - | Not just enterprises and interested parties, but every platform user is a "consumer" | Shifts away from trying to create the right standard for customers, towards allowing discussions and ideas to come from anyone in the ecosystem | -Large enterprise<br>-SMES<br>-Micro-enterprise<br>-Private entities<br>-Organizations<br>-Institutions |
| Job for consumer: | Facilitate interactions outside of the ANP in order to have a greater exchange of ideas and needs | Allow consumers to interact with experts | Make the search and the information on the platform interactive, for consumers and experts to discuss standards, questions, etc. | -Inform<br>-Research<br>-Procure and license<br>-Store and manage centrally<br>-Read and understand<br>-Apply correctly<br>-Obtain documentation<br>-Monitor and keep up-to-date |
| Problems for consumer: | Black box thinking is reduced, as everyone can participate in the standard-setting process | Consumers can use standards for free and give instant feedback.<br>Online interaction can reduce direct costs for consumers | High number of processes, advantages, and disadvantages of each process must be transparent | -Quantity hardly manageable<br>-Costs for the users |
| Gain for consumer: | - | Feedback through community, as well as feedback on standards | Feedback through community, as well as feedback on standards | -Reduction of trade barriers<br>-Increased security<br>-Clarity about the properties of a product and service<br>-Definition of interfaces and compatibility requirements<br>-Demand through greater consumer confidence and acceptance<br>-Simplified order processing and procedures<br>-Feedback on standards |
| Filter for consumer: | Eliminate physical info points | No limitation of reading | Instead of reducing those that read and browse through standards, everyone that is registered can read for free | -Available online<br>-24/7<br>-No charge for reading |
| Transaction for consumer: | Pay per company size, not per standard | Pay for access, not for standards | Pay service level, not per standard | -Pay according to company size, format of standard (XML, PDF, etc.), and service level<br>-Obtain access to platform and standards |

**Table 3.** *Cont.*

| Component from DIN e.V. and Current Ecosystem | 1. Transformation: From "Resource Control" to "Resource Orchestration" | 2. Transformation: From "Internal Optimization" to "External Interaction" | 3. Transformation: From "Customer Value" to "Ecosystem Value" | New Component in Digital Platform Business Model for Standardization |
|---|---|---|---|---|
| Channel for consumer: | Eliminate physical info points | One browser-based application that enables direct interaction between users and experts (no ANP needed) | Draft commenting, searching, standards management, and downloading standards all happen in one place | -Online (browser-based and app) |
| Key resources for consumer: | - | - | - | -Financial resources to obtain access<br>-Internet access |
| Key activities for consumer: | Eliminate physical info points | One browser-based application that enables direct interaction between users and experts | Draft commenting, searching, and downloading standards all happen in one place | -Search online and buy<br>-Access "Norm-Entwurfs-Portal"<br>-Chat with other users and experts<br>-Comment and rate standards |
| Provider: | - | - | - | Experts from<br>-Industry<br>-Research<br>-Consumer side<br>-Public authorities |
| Job for provider: | Providers do not have to pay for their input/contribution | The ANP is not necessary, as there is a direct interaction between implementers and providers | Providers can choose which process they want (e.g., DIN, ISO, IEEE, OS) | -Active participation in the standardization committee<br>-Contribute content and technology to standardization projects<br>-Bringing in technical solutions |
| Problems for provider: | - | - | - | -Time expenditure<br>-Lengthy consensus process<br>-Conflicts of interest with other participants<br>-Language barriers<br>- |
| Gain for provider: | - | Experts set up a page about themselves and their interests, and can build up a network and expand their reach as experts | - | -Shaping the content of standards<br>-Direct exchange with other interest groups<br>-Knowledge advantage<br>-Making own company known<br>-Influence, acceptance, and respect from others |
| Filter for provider: | Only experts from legal entities can join | Experts can converse in topic-driven chat rooms, replacing the TCs and WGs. WGs and TCs are formed need-based by the providers | Expert participation is free, as long as they participate and are rated | -Only experts from legal entities-Free membership<br>-Participation as a free expert |

**Table 3.** *Cont.*

| Component from DIN e.V. and Current Ecosystem | 1. Transformation: From "Resource Control" to "Resource Orchestration" | 2. Transformation: From "Internal Optimization" to "External Interaction" | 3. Transformation: From "Customer Value" to "Ecosystem Value" | New Component in Digital Platform Business Model for Standardization |
|---|---|---|---|---|
| Transaction for provider: | Instead of paying for work in one standards committee, pay for a desired standardization process offered by partners (which is mandatory) | Instead of putting more pressure on standards committees to adapt to new trends, allow experts to choose between or create ad hoc committees for each project | Projects do not happen in a "black box", but the process is made transparent online | -Give knowledge and know-how<br>-Obtain access to standardization projects of interest<br>-Pay fee to special process owners (partner, e.g., DIN, IEC, VDI, OS)—choosing a process is mandatory<br>-Obtain standardization process from partner |
| Channel for provider: | Single point of access for all services | Platform offers plug and play document management systems, but partners can offer other types of systems.<br>Chat rooms allow for interaction between users | Single point of access for all services | -One browser-based online access point for commenting, reading, writing, chatting, discussing |
| Key resources for provider: | Choose and pay for process type offered by partners | Choose and pay for process type offered by partners | Membership is free | -Membership of a legal entity<br>-Choose standardization process type<br>-Recognition of the rules of standardization process owner -Direct costs (process costs, personnel, material, training costs, etc.) |
| Key activities for provider: | - | Set up an account, contribute to standardization projects, discuss with others | Engage with other experts and become known | -Authorization<br>-Granting of copyright usage rights<br>-Declaration of confidentiality<br>-Get to know documents and research options<br>-Acquire knowledge of standardization |
| Partner: | Not only DIN can set standards, but other SSO as well | DIN e.V. does not have to offer various processes, but various SSOs can offer different processes | Offer the best standard-setting experience, by having many competing partners | DIN e.V. |
| Job for partner: | Instead of managing a certain amount of TCs, offer an approved process to all experts.<br>Partners may offer standards in different formats (XML, Source Code, Text, etc.)<br>Partners may offer training | Facilitate discussions and consensus-building among experts | Instead of solely involving TC members (apart from the commenting phase), standardization project includes every expert that contributes money and ideas to be a part of it | -Project management<br>-Workflow management<br>-Document management<br>-Consensus-building<br>-Digital formats<br>-Training |
| Problems for partner: | Higher competition challenge to manage more experts and their opinions than so far | Increased competition | -Challenge to anticipate the smoothness of a project, as more different experts become involved | -Competition with other formal standards organizations and informal standardization activities<br>-The convergence of topics and innovation cycles<br>-Zero error tolerance |

**Table 3.** *Cont.*

| Component from DIN e.V. and Current Ecosystem | 1. Transformation: From "Resource Control" to "Resource Orchestration" | 2. Transformation: From "Internal Optimization" to "External Interaction" | 3. Transformation: From "Customer Value" to "Ecosystem Value" | New Component in Digital Platform Business Model for Standardization |
|---|---|---|---|---|
| Gain for partner: | Access to more experts, grow brand | Access to more experts, grow brand | Generate more revenue through greater reach | -Great reach to all experts<br>-Grow in relevance, reach, and significance for the economy |
| Filter for partner: | Instead of one fixed partner, access is given to many partners, offering their services and resources | Partners can access platform and connect with all experts | Users rate the processes after using them. Poorly rated processes or untrustworthy processes can lose access to the platform | -Accredited standardization organization (accredited processes)<br>-Positive ratings |
| Transaction for partner: | Partners receive money from experts, who want to set a standard | - | - | -Receives money from providers<br>-Gives project management and process management services to providers<br>-Pays owner a transaction fee<br>-Obtains access to platform<br>-Gives knowledge and expertise<br>-Learns from market (consumers) |
| Channel for partner: | Browser-based or via app | Shift away from one system to several workflow and document management systems offered by the partners | Easy to use and intuitive app or SaaS. | Workflow and document management system (Livelink) |
| Key resources for partner: | There is no need for their own server or web shop. This is outsourced onto the platform run by the owner | There is no need their own authors, as the content is provided by experts | - | -Established and secure online and offline standardization processes<br>-Specialized employees<br>-Secure document management system<br>-Developers |
| Key activities for partner: | The identification of innovative topics and technology is done by experts and users, but SSOs can also trigger activities. | Advertise own process and benefits The SSO does not have to facilitate the committees anymore; they organize themselves on the platform | - | -Involvement of all interested parties<br>-Standardization process management up to publication<br>-Ensuring uniformity of all "own" technical standards<br>-Advertise own process and benefits<br>-Review of existing standards |
| Owner: | The owner is not an SSO, but an organization that can best bring consumers, providers, and process owners together (such as Beuth) | To increase interaction, an online standardization platform can help to expand the reach, opposed to the classic TCs that are rather exclusive | By shifting from production-centered to a broader interaction in a community, more value is generated through the exchange | Beuth Verlag |
| Job for owner: | Instead of matching demand and offering standards, match user needs and experts | Instead of offering a standard, experts can choose a process to create, amend, or withdraw a standard | Standards are free and are open for search and reading | -Match demanded and offered standards<br>-Distribution of standards<br>-Standards management tools<br>-Accompanying literature<br>-Training |

**Table 3.** *Cont.*

| Component from DIN e.V. and Current Ecosystem | 1. Transformation: From "Resource Control" to "Resource Orchestration" | 2. Transformation: From "Internal Optimization" to "External Interaction" | 3. Transformation: From "Customer Value" to "Ecosystem Value" | New Component in Digital Platform Business Model for Standardization |
|---|---|---|---|---|
| Problems for owner: | DIN e.V. does not have to analyze trends. Consumers and experts are in control of the topics | Instead of efforts to reduce processing times, experts and consumers can choose their suitable processes | By allowing the rating of experts, standards, and topics, the whole community can create more knowledge and input | -Always suitable for industry needs<br>-All national, European, and international norms, standards, and technical regulations<br>-Clarity and high quality in the search function |
| Gain for owner: | Brand is known for processes. Reach of brand becomes much greater Beuth becomes known as a platform for exchange and standard-setting | The brand reach and community size become much greater as the platform size grows | The brand Beuth becomes key in the realm of standard-setting | -Strong brand<br>-Growth of platform |
| Transaction for owner: | Give access to consumer and receive money | Receive money from partners for access to platform | Experts can use free of charge and obtain access for free | -Receives money from consumer for access<br>-Gives access to buyer<br>-Receives transaction fee from partner<br>-Gives access to partner |
| Channels for promotion by the owner: | Enable partners to advertise | Connect to other b2b platforms | Involve consumers and experts via online discussions, challenges, hackathons, events, etc. | -Fairs<br>-Publications<br>-Website<br>-Flyer<br>-Events<br>-Training<br>-Meetings<br>-Other platforms |
| Key resources for owner: | There is no need for own processes and respective employees, as these are provided by the partners | To create interaction, a server is needed to facilitate the platform and its services and applications. Developers are needed | Additional authors | -Employees<br>-Active authors<br>-Web shop<br>-Server<br>-Developers |
| Key activities for owner: | -List and make standards available<br>-Make available digitally formatted content<br>-Software solutions for standards management | Help SSOs, experts, and consumers to interact.<br>Help authors, experts, and consumers to interact | Authors create work accompanying the standards set by the experts, and create additional streams of revenue through a web shop | -List and make standards available<br>-Develop accompanying offers<br>-Technical literature for all important industries and professional groups<br>-Digitally formatted expert content<br>-Software solutions for standards management |
| Governance by owner: | Consumers and experts rate each other | Consumers and experts rate partners | Best-rated are more visible, poorly-rated may be dismissed | -Ranking of experts<br>-Ranking of standards<br>-Ranking of partners<br>-Accreditation of partners<br>-Accreditation of experts |
| Core value unit: | - | - | - | Listing of a standard or a standardization project |

### 4.3.1. Transforming the Consumers

Regarding the consumer perspective, the 1. transformation (from "resource control" to "resource orchestration") is to facilitate interactions outside of the ANP in order to reduce control and access limitations to ensure a greater exchange of ideas and needs, to reduce "black box thinking", as everyone can participate in the standard-setting process online, and to eliminate physical info points, as consumers can access the platform online and read standards free of charge.

The 2. transformation (from "internal optimization" to "external interaction") is to address not only enterprises and interested parties, but every platform user who wants to access the online platform. The platform will allow consumers to directly interact with any expert, whereas consumers can read and download standards for free and give instant feedback. Online interaction will reduce direct costs for consumers. Experts will receive broad feedback through the community, as well as direct feedback on standards. There will be no limitation to reading standards. Consumers will rather pay for access according to company size and service levels, and not for standards. Consumers' access channel will be solely one browser-based application that enables direct interaction between users and experts (no ANP needed), as well as direct interaction between users and experts.

The 3. transformation (from "customer value" to "ecosystem value") makes the search and the information on the platform interactive, for consumers and experts to discuss standards, questions, etc. Consumers can choose between different standardization processes offered by the partners, which is not the Beuth Verlag, but SSOs, and, therefore, the advantages and disadvantages of each standardization process must be transparent. Consumers give and receive feedback within the online community and give and receive feedback on standards. As mentioned above, instead of reducing those that read and browse through standards through physical info points, everyone that is registered online can read online for free. At the same time, draft commenting, searching, standards management, and downloading standards take place all in one place, on the online platform.

### 4.3.2. Transforming the Providers

Regarding the providers' perspective, the 1. transformation (from "resource control" to "resource orchestration") providers do not need to pay a membership fee to be able to contribute content, know-how, and technologies. Instead of paying for participation in a technical committee, they pay for a desired standardization process offered by partners (which are SSOs). The choice is mandatory for every standardization project. Providers can enjoy having a single point of access for all services. Remaining unchanged is the fact that only experts from legal entities can join.

The 2. transformation (from "internal optimization" to "external interaction") has the effect that the ANP is not necessary anymore, as there is a direct interaction between consumers and providers. Experts set up a publicly visible profile page on the platform, containing information about themselves and their interests. This will help to build a network and expand their reach as experts on the platform. Experts interact in topic-driven chat rooms, eliminating the need for fixed TCs and WGs. TCs and subgroups are formed based on needs by the providers, which, instead of putting more pressure on standards committees to adapt to new trends, allows experts to choose between or create ad hoc committees for each project of interest. The platform offers a plug and play document management system, but partners can offer other types of connected management and exchange systems depending on their used format (XML, Source Code, PDF, etc.).

The 3. transformation (from "customer value" to "ecosystem value") has the effect that providers can choose online, within the platform, which process they want (e.g., DIN, ISO, IEEE, OS). Expert participation is free, as long as they participate and are rated positively. Projects no longer take place in a "black box", as the process is transparent online. Providers enjoy a single point of access for all services, and the membership is free. Experts engage with other experts and build up community merit, similar to Open Source

communities [109]. For a trustworthy platform, poorly-rated or untrustworthy providers can lose access to the platform.

### 4.3.3. Transforming the Partners

Regarding the partner perspective, the 1. transformation (from "resource control" to "resource orchestration") is to switch the role of DIN e.V. and Beuth Verlag. This is a very interesting tweak, as it unlocks greater potential in reach, interaction, and value creation for the online standardization platform. Instead of one fixed partner, access is given to many partners, offering their services and resources; thus, it is not only DIN that facilitates standard-setting processes on the platform, but other SSOs offer their standard-setting processes as well. Instead of managing a certain amount of TCs, DIN e.V., as well as other SSOs, offer their approved processes to all experts. This allows for partners to offer standard-setting processes in different formats (XML, Source Code, Text, etc.). Some partners may offer training with regard to their processes. This increases competition between the SSOs on the platform. SSOs will have the challenge of managing more experts and their opinions than in the present pipeline business model approach. At the same time, access is granted to more experts, which allows the SSO's brand to grow. Partners receive money from experts, who want to set a standard and choose a process. Partners can access the platform through a browser or via an app. As the platform is entirely online, there is no need for an own server or web shop on the side of the partner. This is outsourced onto the platform run by the owner. Likewise, no resources are needed for the identification of innovative topics and technologies, as this is done by the providers and the consumers. Nevertheless, SSOs can also trigger activities, by approaching experts on the platform.

The 2. transformation (from "internal optimization" to "external interaction") has the effect that DIN e.V. does not have to offer various processes to meet market needs. Rather, various SSOs can offer different complementary or competing processes. SSOs can access the platform free of charge and connect with all experts, facilitating discussions and consensus-building among experts. This allows them to grow their own brand. As a partner, DIN e.V. does not have to facilitate the committees anymore; they organize themselves on the platform. The partner does not need its own authors, as the content is provided by interacting with experts. As partners, SSOs will advertise their processes on the platform, and communicate the benefits of their processes for experts. The platform enjoys much greater flexibility and a higher interaction rate, as there is not a limited amount of standardization workflows trying to fit to the market, but there are several workflows and compatible document management systems offered by the partners.

The 3. transformation (from "customer value" to "ecosystem value") creates the highest value by offering the best standard-setting experience, by having many competing partners. Instead of solely involving TC members in the standardization process (apart from the commenting phase), a standardization project includes every expert on the platform, who contribute financial resources, and know how to be a part of it. This, of course, increases the challenge of anticipating the smoothness of a project, as more different experts become involved. At the same time, the browser-based and intuitive platform will generate more revenue through greater reach. In order to ensure high value, experts rate the processes after using them. Poorly-rated or untrustworthy processes can lose access to the platform.

### 4.3.4. Transforming the Owner

Regarding the owner perspective, the 1. transformation (from "resource control" to "resource orchestration") is to switch the role of DIN e.V. and Beuth Verlag. The owner of the platform is not an SSO, but an entity that can bring consumers, providers, and partners together (here, Beuth Verlag). Instead of matching demand to rather static organizational structures and processes, the owner matches user needs, experts, and a variety of processes and output formats online. Consumers and providers are in control of their topics and activities. As the owner, Beuth becomes known as a platform for easy access to, an open community for, and the user-centered setting of standards. The owner facilitates access to

consumers and receives money, which is a shift from the status quo. The owner enables partners to advertise on the platform. Thus, there is no need for their own standardization processes and respective employees, as these are provided by the partners.

The 2. transformation (from "internal optimization" to "external interaction") to an online standardization platform can help to expand the reach and increase interaction, as opposed to the classic TCs, which are rather exclusive. Instead of efforts by the owner to reduce processing times, as is done today, experts and consumers can choose their suitable processes among platform partners. They can choose a process to create, amend, or withdraw a standard. The brand reach and community size become much greater as the platform size grows. For further growth, the platform may connect to other b2b platforms. In order to create interaction, a server is needed to facilitate the platform and its services and applications. The platform helps SSOs, experts, and consumers to interact, as well as authors, experts, and consumers to interact. Consumers, providers, and partners rate each other. What remains unchanged is that the owner receives money from the partners, and grants access to the platform.

Considering the 3. transformation (from "customer value" to "ecosystem value"), by shifting the owner's perspective from a production-centered approach to a broader interaction in a community, more value is generated through the exchange. Standards are free and are open for search and reading. By allowing the rating of experts, standards, and topics, the whole community can generate more feedback and insights, and, thus, create more knowledge and input. The brand "Beuth" becomes key in the realm of standard-setting. Experts can use standards free of charge and obtain access for the exchange of active participation in standardization processes. The owner will make efforts to involve consumers and experts in online discussions, challenges, hackathons, events, etc. Additional authors can create work accompanying the standards set by the experts, and create additional streams of revenue through a web shop for the owner. For the best value, the best-rated standards, experts, and processes are more visible, whereas those that are poorly-rated may be dismissed.

### 4.4. The Standardization Platform Business Model

This chapter describes the platform business model for standardization after transforming its components according to Alstyne et al. (see Figure 8). Consumers on the standardization platform are now large enterprises, SMES, micro-enterprise, private entities, organizations, and institutions. They can inform themselves; research, download, store, and manage centrally and online; read, understand, and apply standards correctly; and obtain documentation and monitor activities on the platform and keep up-to-date. They are confronted with the challenge of managing the large quantities of standards available and costs to access the platform. Benefits for consumers are the reduction of trade barriers, increased security of services and products, clarity about the properties of a product or service, definition of interfaces and compatibility requirements, higher demand by consumers through greater consumer confidence and acceptance, simplified procurement processing and procedures, as well as the possibility to directly offer feedback on standards. The platform is available online 24/7 and, as mentioned above, with regard to transactions, consumers pay an access fee (a sort of flat rate) to the owner according to company size service level (e.g., advanced formats of standards, such as XML, Source Code, etc.). In return, they receive access to the platform and can read all standards free of charge. Consumers access the platform online (browser-based or app). Consumers need to invest financial resources to obtain access, and they need to have internet access. Key activities for the consumer are to log in, create an account, leave payment data, search online, and download the standard; they can access and comment on drafts, chat with other users and experts, as well as comment and rate standards, experts, or processes (offered by the partners).

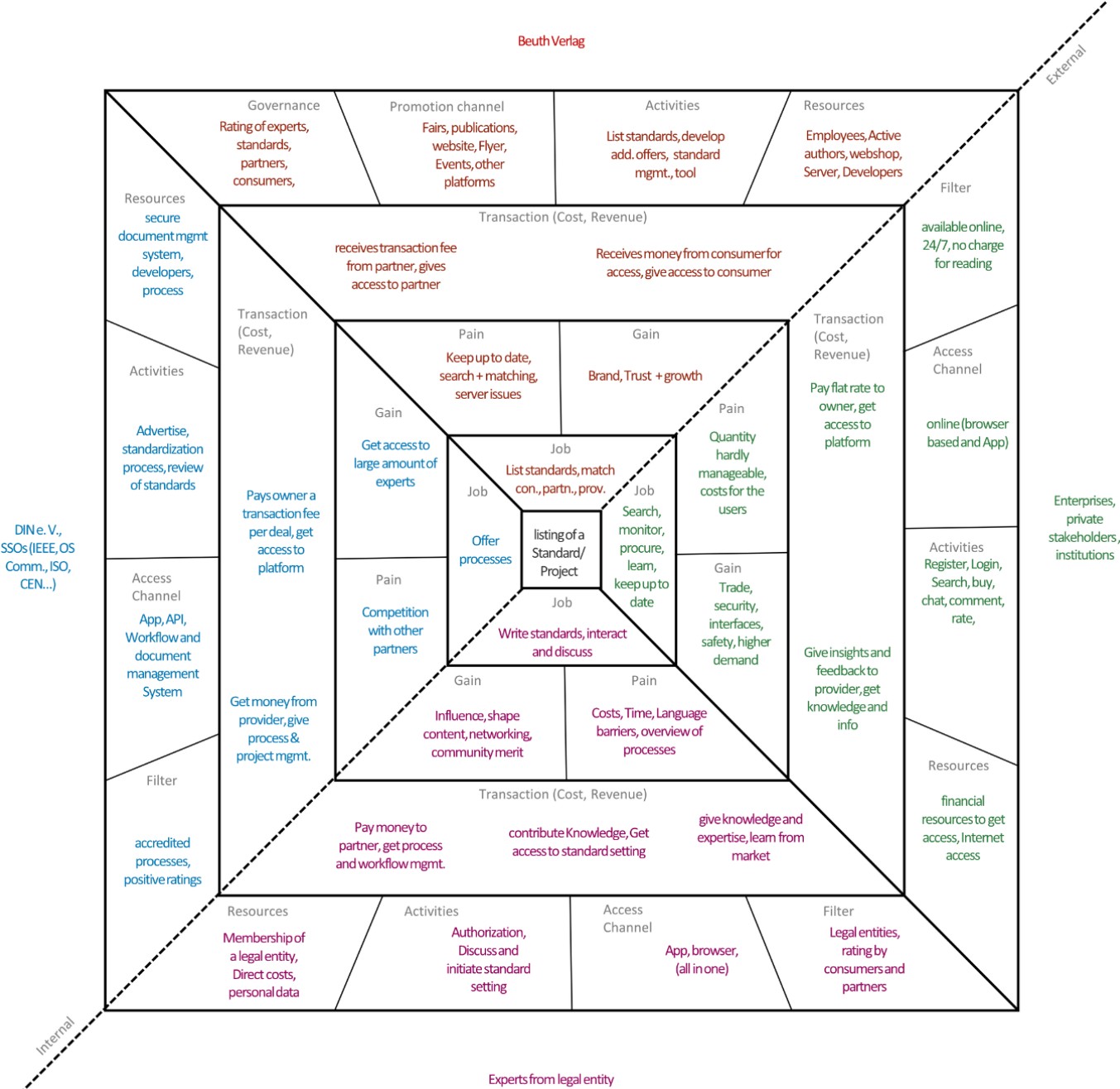

**Figure 8.** The digital standardization platform illustrated via the Platform Business Model Canvas (according to Eisape [55]).

Providers on the standardization platform are experts from industry, research, consumer side, or public authorities. They actively participate in standard-setting processes and discussions online, by contributing content know-how and technology. Their challenges are time expenditure, membership and participation costs (e.g., rising traveling costs), lengthy consensus processes, conflicts of interest with other participants, and sometimes even language barriers. On the other hand, they benefit by shaping the content of standards, by having direct exchange with other interest groups, by gaining a knowledge advantage, by making their own company known, and by gaining influence, as well as acceptance and respect from other experts and consumers (community merit). Only experts from legal entities can be providers and they enjoy free membership. When a provider does not engage in standardization activities over a significant amount of time, the expert's

profile can be switched to a consumer profile. With regard to the transaction, experts offer their knowledge and know-how, and, in return, receive access to standardization projects of interest. They have to pay a fee to a platform partner, who are standardization process facilitators (e.g., DIN, IEC, VDI, OS). Choosing a process is mandatory as soon as a standardization process is to be initiated. Experts share their knowledge and expertise, and, in return, learn from the market (the consumers). Experts only have one browser- or app-based online access point for commenting, reading, writing, chatting, discussing, and sharing on the platform. Providers need to invest key resources in the sense that they need to have membership of a legal entity, choose standardization process types, recognize the rules of respective standardization process owners, and take care of direct costs (e.g., process costs, personnel, material, training costs, etc.). The key activities of providers consist of setting up an account, initiating or contributing to standardization projects, and discussing with others. Further mandatory activities to be able to participate on the platform are the completion of an authorization process, the granting of the copyright usage rights to DIN e.V., the signing of the declaration of confidentiality, learning about documents and research options, and acquiring knowledge on standardization processes.

Partners on the standardization platform are SSOs, such as DIN e.V. They offer project management, workflow management, document management, consensus-building processes, digital formats, and training to experts. Their main challenge is that they are in competition with other formal standards organizations on the platform and perhaps also with informal standardization activities (offline). The convergence of topics and the shortening of innovation cycles will also spark optimization activities within the SSOs to meet market needs. Zero error tolerance also remains a challenge. As platform partners, SSOs gain greater reach to all experts, growing in relevance, reach, and significance for the economy beyond national borders. The platform will only give access to partners that are standardization organizations with tested and approved processes. Their document management and exchange systems need to be compatible, and partners need to achieve positive ratings. On an online standardization platform, partners receive money from providers and offer project management and process management services in return. Partners pay the owner a transaction fee for each standardization project, and, in return, obtain access to the platform, as well as data on topics, processes, activities, and trends. Access channels for partners are APIs (for compatible workflow and document management systems) and the app and browser-based access. Partners need to invest key resources in order to facilitate established and secure online and offline standardization processes, and secure document management systems with specialized employees and developers. Key activities of partners are, depending on the rules of the standardization process, the involvement of all interested parties, the standardization process management up to publication, ensuring uniformity of all their "own" technical standards, advertising their own processes and benefits, and the review of existing standards.

The owner of the platform could be Beuth Verlag. The owner matches users and experts, facilitates the distribution of standards on the platform, and offers standards management tools, as well as chat functions, analysis tools, and accompanying literature. The owner faces the challenge that the platform needs to be relevant, trusted, and beneficial for the users. It has to always be suitable for industry needs; include all national, European, and international standards and technical regulations and specifications; and ensure higher usability, clarity, and high quality in the search function. The advantage is that it can grow a very strong brand and grow the platform. The owner receives money from consumers for access. The owner offers the experts active participation on the platform by offering free access in return. Moreover, it receives transaction fees from the partners, giving them access in return as well. The owner promotes its services through fairs, regular publications, on the platform, flyers, events, and other platforms. Key resources for the owner are employees, active authors, advertising costs, web shops, online tools, servers, and developers. Key activities for the owner are to list standards (e.g., newly initiated, completed, or historic) on the online platform, make standards available for everyone, develop accompanying offers

(e.g., technical literature for all important industries and professional groups), support digitally formatted contents and offer software solutions for standards management, etc. For a healthy platform, the owner introduces governance rules that support the rating of experts, standards, partners, processes, literature, etc. The owner also makes sure to accredit experts and partners in order to grant them access.

The core value unit (which is to be understood as the minimal technical common denominator of all four ecosystem perspectives) is the listing of a standardization project or a completed technical specification (standard). This can, in addition, be connected to a related offer (literature, training, information).

### 4.5. Comparing the Two Ecosystems via Index

Comparing the two ecosystems via the platform index helps to obtain insight into how much more potential lies in the new digital platform in contrast to the current business ecosystem (see Table 4). The comparison shows that such a platform could disrupt the entire standardization industry (see Figure 9). The revenue potential of Beuth is twice the value of DIN e.V. (78 Mn. € compared to 40 Mn. €). The number of customers from Beuth is around 168,000. DIN has fewer customers, as not all Beuth customers buy DIN's standards. Moreover, the number of experts is no match. Beuth sells the output of experts from SSOs worldwide, whereas DIN is limited to its resource of 32,000 German experts. Looking at the number of partners, DIN, in the core of its ecosystem, works with one partner (Beuth), whereas Beuth works with approx. 80 SSOs. Comparing the brands in a Google ranking check with regard to the keyword "Normen" (English: standard), DIN e.V. surpasses Beuth by ranking third compared to fifth. The number of core value units is key for a successful platform, and 34,000 standards versus 600,000 give Beuth a clear edge. Furthermore, the auxiliary value units, which are products offered by the partners, allow Beuth to stand out, with 100 partner products, whereas DIN only has six.

The index indicates that the standardization platform with an index of 0.77 is likely to have far more potential to be a successful platform than the pipeline business model and the core ecosystem of DIN e.V. (index: 0.09).

**Table 4.** Comparing DIN e.V. ecosystem versus the developed standardization platform concepts via the two platform indexes (1 and 2).

| Platform KPIs According to Eisape [87] | Platform KPIs Used in this Research, Due to Availability of Data | DIN e.V. Ecosystem (Version 1) | # | Standardization Platform (Version 2) | # |
|---|---|---|---|---|---|
| Revenue value and diversity | Owner revenue | 40 million € [148] | 1 | 78 million € [118] | 5 |
| Market share of consumer | Numbers of customers (users of standards) | Less than 168,000 [118] | 2 | Approx. 168,000 [118] | 4 |
| Market share of provider | Number of providers (experts) | 32,000 (only national experts) [149] | 1 | All national and international experts [118] | 5 |
| Number of partnerships | Number of partners (pubvialishers/SSOs) | 1 publisher [134] | 1 | 80 [150] | 5 |
| Brand value of owner | Website rankings | 3rd place for "DIN.de" [151] | 4 | Ranked 5th [151] | 2 |
| Number of core value units | Number of available standards | 34,000 [149] | 1 | 600,000 [152] | 5 |
| Number of auxiliary value units | Number of product types by the partner | 6 [152] | 1 | 100 [150] | 5 |
| Index | | DIN e.V. ecosystem | 0.09 | Digital platform | 0.77 |

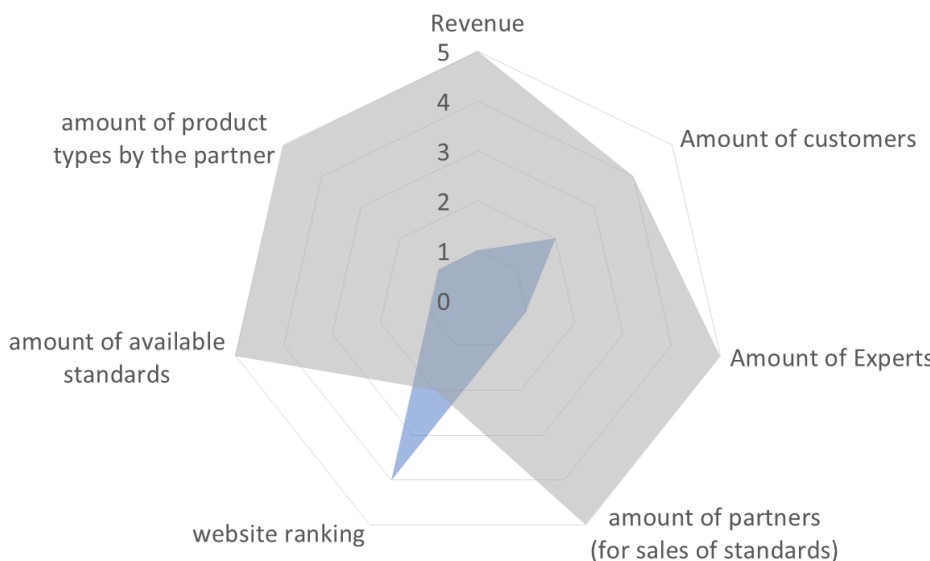

**Figure 9.** Comparing DIN's ecosystem with the developed digital standardization platform (adapted from Eisape [97]).

## 5. Discussion

The two-step approach introduced in this paper according to the design science framework offers a theoretical and practical model for researchers and practitioners to design and anticipate a possible business model for digital platforms in their industries. The model was derived from the literature, with consideration of the digital evolution of the music industry. The model's adequacy was demonstrated via the illustrative case study of the German standards-setting organization (DIN e.V.) and its ecosystem. Consequently, this study mapped DIN's business model and ecosystem with the Platform Business Model Canvas, and then transformed it into a digital platform. The results show several aspects that are very interesting for practitioners and researchers.

Regarding the field of standardization, this research shows that, although SSOs digitize their processes, products, and services, and although they also consider online standardization approaches, they are still hesitant and leave a void that can be filled by a smart market player, disrupting the entire industry. Digitizing the products and processes is not enough. Transforming their own business model is risky, as is having little understanding of how digital transformation can disrupt the entire ecosystem [48–50]. Tools needed to anticipate digital platforms in an ecosystem are difficult to find and often remain in the theoretical realm. The results from this study take a step towards filling this gap, and present a practical approach.

DIN e.V. may claim to be a "platform" for standardization in Germany and worldwide, as it offers experts and companies the possibility to influence standards and regulations on a national and international level. However, with regard to the understanding of a platform as a two-sided market, the business model of a standard-setting organization in general, and DIN e.V. in particular, resembles pipeline businesses that seek to optimize their value chain through digitization, without affecting their known importance and expertise in the world of standardization. The results show that transforming the business model and the ecosystem of DIN e.V. into a digital platform creates entirely new potential. The platform index of the digital ecosystem—and, hence, the platform—is 8,5 times greater than the platform index of the entire current ecosystem of DIN and its partners. Another finding of this study is that, compared to the current ecosystem setup, where DIN is essentially the central player, in a digital platform, a player such as DIN's publishing partner and

subsidiary company, Beuth Verlag, could be at the center of matching demand and offer, which could possess the potential to become a disruptive force in the standardization industry. DIN e.V. would risk becoming just one of many standard-setting organizations and lose relevance, as Beuth Verlag could achieve much greater market relevance and foster much shorter reaction times regarding innovation cycles. Therefore, anticipating the digital transformation of one's ecosystem could be vital for standard-setting organizations to develop a platform strategy.

### 5.1. Transformation of a Traditional Pipeline Business Model, and Its Relation with Open Innovation

Open innovation, together with the digitization of products, processes, and services, can result in the digital transformation of an existing pipeline business model towards a digital platform. Against the background that platform business models can develop an enormous disruptive character, the conceptional derivation of a digital platform business model that takes into account the actors of the incumbent ecosystem anticipate a possible business model innovation scenario. Due to their open character, platforms create enormous growth in almost every industry. An early consideration of such a scenario can, thus, help to contribute innovative ideas to one's own innovation efforts (e.g., in the context of an open innovation approach). This paper introduces an approach to conceptualize the emergence of a platform business model in any given industry. Further research on how the anticipated conceptual platform business model helps to generate further innovative ideas in an open innovation process can contribute further insights.

### 5.2. Future Research

This paper introduces an approach to conceptually transform pipeline business models into platform business models in two steps. This is performed by using the PBMC to map a pipeline business model and its core ecosystem, and subsequently transforming the business model components along the three transformation points of Alstyne et al. to anticipate a digital platform business model. This theoretic approach helps develop a conceptual scenario and a strategic understanding on what a platform business model in a specific industry setting may look like.

Further research could generate a deeper understanding on how to anticipate digital transformation on a business model level. Further case studies would help compare the anticipated platform business model not only to the pipeline business models from which it emerged, but also to existing, comparable platform business models.

Future research could anticipate more conceptual platform business model versions, where a respective pipeline business occupies different roles (owner, partner, consumer, contributor).

Governments, which play a vital role in the standardization industry, were not considered in this paper. Making the government a key player of a standardization platform (e.g., the owner) could lead to another very interesting conceptual platform business model variation.

Furthermore, the mapping was carried out with the PBMC introduced by Eisape. The use of another Platform Business Model Canvas could result in an entirely different platform business model; thus, this paper proposes only one way of anticipating digital transformation. Further research could compare different approaches.

The employed platform index helps to compare the theoretical potential of two platform business models. Generating the platform index does not require special knowledge, as it can be employed with available market data and even estimates. Its adequacy was demonstrated in this paper, but further research is needed to gain more insights and confirm the results of this paper. Further research would need to elaborate how appropriate the "judgement" of the index is compared to reality. Another interesting approach would be to compare not two, but a multitude of platforms via the index, and further evaluate them to additional market data. Further research on the correlation of the platform index and the real success of platforms will support future results regarding this approach.

*5.3. Limitations*

The results of this paper are subject to certain limitations, which are outlined in the following.

The results here are based on literature review and one illustrative case study. The same approach of mapping the pipeline business model and the ecosystem, but then subsequently discussing the transformation possibilities with the strategic management of DIN e.V. in interviews or workshop formats, could lead to an entirely different platform business model with perhaps an even higher platform index or a more feasible platform business model.

The developed platform business model is a theoretical one. Framework conditions or market constellations possibly opposing the implementation of such a platform were not analyzed. To the critical reader, it might seem farfetched to declare the created platform business model for standardization a disruptive force, which is yet to be proven. Nevertheless, given the greater reach and the larger base of consumers and providers in the new digital ecosystem, the potential for a disruptive business model is shown in theory and expressed through the index.

A critical aspect with regard to the comparison is the availability and choice of data. Depending on which data are available and used, the result of the comparison may vary significantly. This could be a subject for further research to identify the right set of data for comparison for each KPI. In this paper, the data used came from the websites of the ecosystem players, and represent the incumbent market situation. This means that the comparison was not performed with fictive data, but with current data, as if the data would apply to the platform today. Of course, this neglects possible effects (network effects, implementation risks, dynamic competition, etc.) that would alter the data significantly. However, it offers mutual grounds to compare two platform business model concepts according to Eisape, as well as a starting point for business model innovation.

**6. Conclusions**

Following the design science framework, this paper introduced a two-step approach to transform a pipeline business model (e.g., of SSO) to a conceptual digital (standardization) platform. This was performed by mapping the current pipeline business models and its ecosystem with the Platform Business Model Canvas introduced by Eisape. The representation of the current ecosystem was then digitally transformed according to the three key transformation points introduced by Alstyne et al., conceptually shifting the current ecosystem into the digital realm. The illustrative case study on DIN e.V. (the German standardization organization) demonstrated and evaluated the adequacy of this new approach.

The results of this paper directly correspond to the scientific discussion in the literature on how pipeline businesses can anticipate the digital transformation of their pipeline business model and their ecosystem into a digital platform. Secondly, this research allows practitioners to use the developed tool in their platform strategy, for business model innovation, and for the strategic anticipation of a disruptive business model in their ecosystem.

**Funding:** This research received no external funding.

**Institutional Review Board Statement:** Not applicable.

**Informed Consent Statement:** Not applicable.

**Data Availability Statement:** All sources of the data used in this article are listed under references.

**Conflicts of Interest:** The author declares no conflict of interest.

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
