# Peer review of "Transforming Pipelines into Digital Platforms: An Illustrative Case Study Transforming a Traditional Pipeline Business Model in the Standardization Industry into a Digital Platform"

_2199-8531, doi:10.3390/joitmc8040183_

Round 1

Reviewer 1 Report

The value of the article lies in demonstrating the practical application of tools for transforming and predicting the development of digital platforms in the enterprise ecosystem. The author based on the approaches of Schallmo et al. and Alstyne et al. propose an approach to the digital transformation of business processes of enterprises, and also demonstrates this approach in detail using the example of two enterprises from different business areas, describes the features and benefits of the transformation results obtained, and presents a methodology for quantifying the potential of the former and transformed business model (digital platform index).

 • Is the manuscript clear, relevant to the field, and presented in a well-structured manner?

The authors consider the topical issue of digital transformation of enterprises. The presented manuscript is well structured and follows the accepted structure for articles in the journal.

• Are the references cited mostly recent (within the last 5 years) and up-to-date? Does it include excessive self-citations?

 All sources used are relevant and applicable to the manuscript.

• Is the manuscript scientifically sound and is the experimental design suitable for testing the hypothesis?

The methods and tools used by the authors are scientifically substantiated. The sequence of applying the theoretical concepts of business model transformation is adequate to the goal of the manuscript. The developed approach to transformation for enterprises is checked for adequacy on the example of existing enterprises.

• Are the results of the manuscript reproducible based on the details provided in the Methods section?

The results of the manuscript correspond to the theoretical approaches and tools of the Methods section.

• Are the figures/tables/images/diagrams appropriate? Are they displaying the data correctly? Are they easy to interpret and understand? Is the data interpreted correctly and consistently throughout the manuscript?

All figures and tables in the manuscript are informative and help to understand the reasoning and conclusions of the authors. However, some tables contain invalid source references.

• Are the conclusions consistent with the evidence and arguments presented?

The conclusions of the manuscript are fully consistent with the evidence and arguments. In addition, the findings consider in detail the limitations of the author's approach to the digital transformation of business processes (which is especially valuable for enterprises) and directions for future research in this area.

I thank the author for the opportunity to get acquainted with your research. I have enjoyed reading your manuscript.

While appreciating your research, I would like to draw your attention to the following remarks:

- it is necessary to correct invalid references to sources in table 1 and line 668,

- it is necessary to put a link in the corresponding paragraph to Figure 7,

- in section “2.3.1. Digital Transformation According to Schallmo et al.” when considering the concept of SMART, the authors interpret the abbreviation A as Attractive. In the conventional sense, the abbreviation A is interpreted as Achievable. If this is the author's approach to the consideration of the SMART concept, then I ask you to explain it. If this is a mistake, then it needs to be corrected.

Author Response

Dear reviewer,

I very much appreciate your comments and I attended all issues as follows:

- it is necessary to correct invalid references to sources in table 1 and line 668,

This has been fixed

- it is necessary to put a link in the corresponding paragraph to Figure 7,

This has been fixed

- in section “2.3.1. Digital Transformation According to Schallmo et al.” when considering the concept of SMART, the authors interpret the abbreviation A as Attractive. In the conventional sense, the abbreviation A is interpreted as Achievable. If this is the author's approach to the consideration of the SMART concept, then I ask you to explain it. If this is a mistake, then it needs to be corrected.

This has been fixed

I have also revised some parts in the introduction and the discussion as highlighted in green colour.

Thank you so much for making this paper better with your comments.

Best Wishes

Reviewer 2 Report

Dear Authors, 

Thank you for the opportunity to review your work. I've identified several areas that need further improvement, specifically literature review / theoretical background.

The introduction section needs more work on problematisation. I suggest to remove direct quotes from the introduction section (from 41). Also, outline what is the research question of this study. 

First, please have a look at Bonina et al. (2021) and how they defined the complementors and contributors. 

Bonina, C., Koskinen, K., Eaton, B., & Gawer, A. (2021). Digital platforms for development: Foundations and research agenda. Information Systems Journal31(6), 869-902.

Second, please have a look at Jovanovic et al. (2021) and check how they defined industrial digital platforms as well as platform ecosystems. 

Jovanovic, M., Sjödin, D., & Parida, V. (2021). Co-evolution of platform architecture, platform services, and platform governance: Expanding the platform value of industrial digital platforms. Technovation, 102218.

Third, please have a look at Jovanovic et al. (2022) for further insights in relation to platform regulation, platform bylaws.  

Jovanovic, M., Kostić, N., Sebastian, I. M., & Sedej, T. (2022). Managing a blockchain-based platform ecosystem for industry-wide adoption: The case of TradeLens. Technological Forecasting and Social Change184, 121981.

Fourth, Figure 2. is very hard to follow, consider re-designing. Maybe, make a table out of it.

Fifth, there are several areas in the manuscript with "Error! Reference source not found.". Please fix. 

Sixth, discussion section is very underdeveloped. How this study is contributing back to the platform and business model literature.

I hope this helps. 

Author Response

Dear reviewer,

I very much appreciate your comments and I attended all issues as follows:

 The introduction section needs more work on problematisation. I suggest to remove direct quotes from the introduction section (from 41). Also, outline what is the research question of this study. 

Introduction and research goal has been revised to be more specific. Changes are highlighted in green colour.

First, please have a look at Bonina et al. (2021) and how they defined the complementors and contributors. 

Bonina, C., Koskinen, K., Eaton, B., & Gawer, A. (2021). Digital platforms for development: Foundations and research agenda. Information Systems Journal31(6), 869-902.

I included this reference in the introduction

Second, please have a look at Jovanovic et al. (2021) and check how they defined industrial digital platforms as well as platform ecosystems. 

Jovanovic, M., Sjödin, D., & Parida, V. (2021). Co-evolution of platform architecture, platform services, and platform governance: Expanding the platform value of industrial digital platforms. Technovation, 102218.

I included this reference in the introduction

Third, please have a look at Jovanovic et al. (2022) for further insights in relation to platform regulation, platform bylaws.  

Jovanovic, M., Kostić, N., Sebastian, I. M., & Sedej, T. (2022). Managing a blockchain-based platform ecosystem for industry-wide adoption: The case of TradeLens. Technological Forecasting and Social Change184, 121981.

I included this reference in the introduction

Fourth, Figure 2. is very hard to follow, consider re-designing. Maybe, make a table out of it.

This is a platform business model canvas introduced by Eisape that I reference in this work. It is what the canvas looks like.

Fifth, there are several areas in the manuscript with "Error! Reference source not found.". Please fix. 

This has been fixed

Sixth, discussion section is very underdeveloped. How this study is contributing back to the platform and business model literature.

Discussion has been revised to be more specific

Thank you so much for making this paper better with your comments.

Best Wishes